# On the Bias-Variance-Cost Tradeoff of Stochastic Optimization

**Yifan Hu**[*]
UIUC
`yifanhu3@illinois.edu`

**Xin Chen**[*]
UIUC
`xinchen@illinois.edu`

**Niao He**[†]
ETH Zürich
`niao.he@inf.ethz.ch`

## Abstract

We consider stochastic optimization when one only has access to biased stochastic oracles of the objective, and obtaining stochastic gradients with low biases comes at high costs. This setting captures a variety of optimization paradigms widely used in machine learning, such as conditional stochastic optimization, bilevel optimization, and distributionally robust optimization. We examine a family of multi-level Monte Carlo (MLMC) gradient methods that exploit a delicate trade-off among the bias, the variance, and the oracle cost. We provide a systematic study of their convergences and total computation complexities for strongly convex, convex, and nonconvex objectives, and demonstrate their superiority over the naive biased stochastic gradient method. Moreover, when applied to conditional stochastic optimization, the MLMC gradient methods significantly improve the best-known sample complexity in the literature.

## 1 Introduction

Solving modern machine learning tasks relies heavily on stochastic gradient descent (SGD) and many of its variants. Whilst the vanilla SGD depends crucially on unbiased gradient oracles, constructing unbiased gradient estimators can be very expensive or even impossible for many emerging machine learning applications. Such applications include GANs with regularization [25], distributionally robust optimization [26, 14], conditional stochastic optimization [20, 19], meta-learning [30, 11], min-max optimization [32, 33], and bilevel optimization [16], just to name a few.

As an alternative, one often resorts to some natural biased gradient estimators. Existing work on biased gradient methods [17, 1, 18, 7, 23] focuses on the iteration complexity given biased stochastic oracles while ignoring the possibility of constructing refined stochastic oracles with lower bias at a higher cost, e.g., more samples or more computation power. In this paper, we build a unified framework to study the fundamental tradeoff among the bias, the variance, and the cost.

We study optimization problems of the general form

$$\min_{x \in \mathbb{R}^d} F(x), \tag{1}$$

and assume that one does not have access to unbiased gradient estimator of $F(x)$ via simple Monte Carlo sampling or simulation. Instead we assume that one can construct a sequence of approximations

---

[*]Department of Industrial and Enterprise Systems Engineering, University of Illinois at Urbana-Champaign.
[†]Optimization and Decision Intelligence (ODI) Group, Department of Computer Science, ETH Zürich.

35th Conference on Neural Information Processing Systems (NeurIPS 2021).

of $F(x)$, denoted as $\{F^l(x)\}_{l=0}^\infty$. Unlike the original objective $F(x)$, unbiased function value and gradient estimators of each $F^l(x)$ (correspondingly biased estimators of $F(x)$) are accessible through some oracles. However, the cost for querying such oracles increases as the approximation accuracy increases. We measure the bias level using the approximation error either in term of the function values $|F(x) - F^l(x)|$ or in terms of the gradient $\|\nabla F^l(x) - \nabla F(x)\|_2^2$. For simplicity, throughout the paper, we assume $F$ and $F^l$ are smooth, namely, they have Lipschitz continuous gradients.

More precisely, without loss of generality, we assume that (i) for any given $x$, the function approximation error satisfies $|F^l(x) - F(x)| = \mathcal{O}(2^{-al})$ for some constant $a > 0$ under the convex setting, or the gradient approximation error satisfies $\|\nabla F^l(x) - \nabla F(x)\|_2^2 = \mathcal{O}(2^{-al})$ under the nonconvex setting; (ii) for each $l \in \mathbb{N}$, there exists a stochastic oracle $(\mathcal{SO}^l)$ that returns an unbiased estimator $H^l(x, \zeta^l)$ of $\nabla F^l(x) - \nabla F^{l-1}(x)$ given $x$ with bounded variance: $\mathbb{V}(H^l(x, \zeta^l)) = \mathcal{O}(2^{-bl})$ for some constant $b > 0$; (iii) the cost to query the oracle $\mathcal{SO}^l$ is $C_l = \mathcal{O}(2^{cl})$ for some constant $c > 0$. Assumption (ii) indicates that the variance of the stochastic estimator of the function difference decreases with $l$. Intuitively, as function $F^l$ is close to function $F^{l-1}$, one can construct highly correlated gradient estimators for $\nabla F^l(x)$ and $\nabla F^{l-1}(x)$ such that the variance of their difference is small. As we will show later, these constants $a, b, c$ play a crucial role in the complexity of the problems of our interest.

## 1.1 Motivational Examples

**Conditional Stochastic Optimization (CSO)** In CSO [19], the objective function has the following form: $F(x) := \mathbb{E}_\xi f_\xi(\mathbb{E}_{\eta|\xi} g_\eta(x, \xi))$, where $\xi$ and $\eta$ are random vectors. In general, it is not easy to directly obtain unbiased gradient estimators from samples of $P(\xi)$ and $P(\eta|\xi)$, because of the nonlinearity of $f_\xi(\cdot)$. Instead, we can construct a sequence of approximation functions given by $F^l(x) = \mathbb{E}_{\zeta^l = [\xi, \eta_1, ..., \eta_{2^l}]} \left[ f_\xi \left( \frac{1}{2^l} \sum_{j=1}^{2^l} g_{\eta_j}(x, \xi) \right) \right], l = 0, 1, \ldots$, where $\{\eta_j\}_{j=1}^{2^l}$ are independent and identically distributed (i.i.d.) random vectors from $\mathbb{P}(\eta|\xi)$. Unlike $F(x)$, unbiased estimators for $\nabla F^l(x)$ can be easily obtained directly through samples. Under mild conditions, we can show that assumptions (i)-(iii) hold true; in particular, we have $a = 1, b = 1, c = 1$.

**Distributionally Robust Optimization (DRO)** DRO minimizes the expected loss with respect to the worst distribution in an uncertainty set $\mathcal{U}(P)$ for a given distribution $P$. The objective of DRO has the form: $\min_{x \in \mathcal{X}} F_P(x) := \sup_{Q \in \mathcal{U}(P)} \mathbb{E}_{\xi \sim Q} \ell(x; \xi)$. Because of the maximal operator over the uncertainty set, it is hard to obtain unbiased gradient estimator using samples from $P$. Instead one constructs a series of approximation functions given by $F^l(x) = \sum_{i=1}^{2^l} q_i^* \ell(x; \xi_i)$, where $\{\xi_i\}_{i=1}^{2^l}$ are i.i.d. samples from $P$, $P_n$ denotes the empirical distribution, and $\mathbf{q}^* = \{q_i^*\}_{i=1}^{2^l}$ attains the maximum of $\sup_{\mathbf{q} \in \Delta_{2^l}, \mathbf{q} \in \mathcal{U}(P_n)} \sum_{i=1}^{2^l} q_i \ell(x; \xi_i)$ for a $2^l$-dimensional simplex $\Delta_{2^l}$. For CVaR and $\chi^2$-penalty DRO problems, Levy et al. [26] showed that assumptions (i)-(iii) hold true with $a = 2, b = 1, c = 1$ for CVaR DRO and $a = 1, b = 1, c = 1$ for $\chi^2$-penalized DRO.

## 1.2 Multilevel Monte Carlo (MLMC) Techniques

A naive approach to solve (1) is to perform SGD on the approximation function $F^L(x)$ with a level $L$ such that the bias is small and within a targeted accuracy $\epsilon$. We call such a framework $L$-SGD. This encapsulates several existing algorithms under different contexts, such as BSGD [20] in conditional stochastic optimization, multi-step MAML [21] in meta-learning, and biased gradient method [26] in distributionally robust optimization. Despite its simplicity, $L$-SGD generally fails to achieve the smallest total cost since it requires expensive stochastic oracles to ensure small bias.

On the other hand, the *multi-level Monte Carlo* (MLMC) sampling technique, originally designed for stochastic simulation [15], is amenable to obtain better gradient estimators by exploiting the bias-cost tradeoff. The idea is to query fewer oracles with higher cost and smaller bias, and more oracles with lower cost and larger bias; thus, the cost can be effectively reduced. There exist several variations of MLMC with randomization, including the MLMC estimator with randomized truncation (denoted as RT-MLMC) [5], the MLMC estimator with importance sampling (denoted RU-MLMC), the Russian roulette estimator (denoted as RR-MLMC) that uses randomized telescope [22].

Table 1: Summary of the total costs of the algorithms for finding $\epsilon$-optimal solution for (strongly) convex $F$ and $\epsilon$-stationary point for nonconvex smooth $F$. Here $a$ is the decrease rate in the bias, $b$ is the decrease rate of the variance, $c$ is the increase rate of oracle cost, and $\widetilde{\mathcal{O}}(\cdot)$ represents the order hiding logarithmic factors. N.A. stands for not applicable. Let $n_1 = \max\{1, c/a\}$, $n_2 = \max\{1 + (c-b)/a, c/a\}$.

| Algorithms | **V-MLMC** | **RT-MLMC** | **RU-MLMC** **RR-MLMC** | **$L$-SGD** | Convexity |
|---|---|---|---|---|---|
| Bias | $\mathcal{O}(\epsilon)$ | $\mathcal{O}(\epsilon)$ | $0$ | $\mathcal{O}(\epsilon)$ | |
| Mini-batch | Yes | No | No | No | |
| Variance when $c<b$ | $\mathcal{O}(\epsilon)$ | $\mathcal{O}(1)$ | $\mathcal{O}(1)$ | $\mathcal{O}(1)$ | |
| Per-iter cost when $c<b$ | $\mathcal{O}(\epsilon^{-1})$ | $\mathcal{O}(1)$ | $\mathcal{O}(1)$ | $\mathcal{O}(\epsilon^{-c/a})$ | |
| Total Cost if $c<b$ | $\widetilde{\mathcal{O}}(\epsilon^{-n_1})$ | $\mathcal{O}(\epsilon^{-1})$ | $\mathcal{O}(\epsilon^{-1})$ | $\mathcal{O}(\epsilon^{-1-c/a})$ | Strongly Convex |
| | $\mathcal{O}(\epsilon^{-1-n_1})$ | $\mathcal{O}(\epsilon^{-2})$ | $\mathcal{O}(\epsilon^{-2})$ | $\mathcal{O}(\epsilon^{-2-c/a})$ | Convex |
| | $\mathcal{O}(\epsilon^{-2-2n_1})$ | $\mathcal{O}(\epsilon^{-4})$ | $\mathcal{O}(\epsilon^{-4})$ | $\mathcal{O}(\epsilon^{-4-2c/a})$ | Nonconvex |
| Total Cost if $c \geq b$ | $\widetilde{\mathcal{O}}(\epsilon^{-n_2})$ | $\widetilde{\mathcal{O}}(\epsilon^{-1-(c-b)/a})$ | N.A. | $\mathcal{O}(\epsilon^{-1-c/a})$ | Strongly Convex |
| | $\widetilde{\mathcal{O}}(\epsilon^{-1-n_2})$ | $\widetilde{\mathcal{O}}(\epsilon^{-2-(c-b)/a})$ | N.A. | $\mathcal{O}(\epsilon^{-2-c/a})$ | Convex |
| | $\widetilde{\mathcal{O}}(\epsilon^{-2-2n_2})$ | $\widetilde{\mathcal{O}}(\epsilon^{-4-2(c-b)/a})$ | N.A. | $\mathcal{O}(\epsilon^{-4-2c/a})$ | Nonconvex |

These MLMC techniques have been recently explored in different optimization contexts. For instance, Blanchet and Glynn [5] combined the MLMC idea with sample average approximation to address stochastic optimization. Dereich and Müller-Gronbach [10] integrated the vanilla MLMC (denoted as V-MLMC) gradient estimator with Robbins-Monro and Polyak-Ruppert stochastic approximation schemes, and analyzed their convergences. Recently, the RT-MLMC technique was applied to stochastic compositional optimization with strongly convex objectives in Blanchet et al. [4] and distributional robust optimization in Levy et al. [26], Ghosh and Squillante [14]. Beatson and Adams [3] studied the asymptotic behaviors of stochastic gradient descent with RT-MLMC and RR-MLMC estimators. Concurrently, Asi et al. [2] applied RT-MLMC to estimating proximal points and Moreau-Yoshida envelope gradient, minimization of maximal loss, and proximal gradient methods.

## 1.3 Our Contributions

In this paper, we provide a systematic comparison among these MLMC techniques when combined with stochastic gradient descent. Our primary focus is on the analysis of their total computation cost for achieving an $\epsilon$-optimal solution for convex problems and an $\epsilon$-stationary point for nonconvex problems, which is largely missing in the literature. Our contributions are three-fold:

**Non-asymptotic analysis of MLMC gradient methods.** We analyze the non-asymptotic performances and computation complexities of various MLMC gradient methods under the general stochastic optimization framework in all regimes, e.g., different convexity assumptions and different combinations of $a, b, c$. Our main results are summarized in Table 1. Previous results either focus on specific applications [4, 26] or only asymptotic behaviors [3] in restricted regimes.

**Theoretical comparisons and new insights.** Our comparative study of different MLMC techniques yields several interesting findings: (1) When $b > c$, i.e., the variance decays faster than the increase of the cost, all three randomized MLMC gradient methods and V-MLMC (only when $a \geq \min\{b, c\}$) nearly match the complexity of classical unbiased SGD, implying that the problem is no harder than classical stochastic optimization, regardless of the bias. (2) When $b \leq c$, the unbiased MLMC constructions, RU-MLMC and RR-MLMC, are no longer applicable since either the expected per-iteration cost or the variance of the gradient estimator goes to infinity. In this case, RT-MLMC is the most favorable approach without requiring a mini-batch. (3) In all regimes with $0 < a < \infty$ (i.e., bias exists), using the naive $L$-SGD is strictly sub-optimal in terms of the total cost, demonstrating the importance of the exploitation of the bias-variance-cost tradeoff.

**Improved sample complexity for conditional stochastic optimization.** When applied to CSO problems (resp. $a = b = c = 1$), we show that the RT-MLMC and the vanilla MLMC gradient methods significantly improve over existing results reported in the literature. In particular, for nonconvex

smooth CSO problems, RT-MLMC and vanilla MLMC achieve $\widetilde{\mathcal{O}}(\epsilon^{-4})$ sample complexity, which is even better than the best-known $\widetilde{\mathcal{O}}(\epsilon^{-5})$ achieved by the variance reduction technique [20].

## 1.4 Preliminaries

A function $f(\cdot) : \mathbb{R}^d \to \mathbb{R}$ is $L$-Lipschitz continuous if $|f(x) - f(y)| \leq L\|x - y\|_2$ holds for any $x, y \in \mathbb{R}^d$. A function $f(\cdot)$ is $S$-smooth on $\mathbb{R}^d$, if it is continuously differentiable on $\mathbb{R}^d$ and it holds that $\|\nabla f(x) - \nabla f(y)\|_2 \leq S\|x - y\|_2$ for any $x, y \in \mathbb{R}^d$. For any $x, y \in \mathbb{R}^d$, and a continuously differentiable function $f$, if it holds that $f(x) - f(y) - \nabla f(y)^\top (x - y) \geq \frac{\mu}{2}\|x - y\|_2^2$, we say $f$ is $\mu$-strongly convex when $\mu > 0$ and $f$ is convex when $\mu = 0$. Let $x^* \in \arg\min_x F(x)$. We say $x$ is an $\epsilon$-optimal solution if $F(x) - F(x^*) \leq \epsilon$, and $x$ is an $\epsilon$-stationarity point of $F$ if $\|\nabla F(x)\|_2^2 \leq \epsilon$. Polyak-Łojasiewicz (PL) condition [24] is a generalization of strong convexity such that $\|\nabla F(x)\|_2^2 \geq 2\mu(F(x) - F(x^*))$ for $\mu > 0$. By convention we let $\nabla F^{-1}(x) = 0$. For simplicity, throughout the paper, we assume the desired accuracy $\epsilon > 0$ is small enough so that $\epsilon^{-1} \geq \log(\epsilon^{-1}) \geq 1$. We use $\mathcal{O}(\cdot)$ to hide constants that does not depend on the desired accuracy $\epsilon$. $\widetilde{\mathcal{O}}$ further hides the $\log(\epsilon^{-1})$ term.

## 2 MLMC Gradient Methods

### 2.1 Assumptions

We first formally restate the problem setting and assumptions we use throughout the paper.

**Assumption 2.1.** *We assume that there exist constants $a, b, c, M_a, M_b, M_c, \sigma > 0$ such that for any $x \in \mathbb{R}^d$ and $l \in \mathbb{N}$, the following hold.*

(a). *The function value approximation error is bounded: $|F^l(x) - F(x)| \leq B_l := M_a 2^{-al}$.*

(b). *There exits a stochastic oracle, $\mathcal{SO}^l$, that for given $x$ returns stochastic estimators $h^l(x, \zeta^l)$ and $H^l(x, \zeta^l)$ such that*

$$\mathbb{E}h^l(x, \zeta^l) = \nabla F^l(x); \mathbb{V}(h^l(x, \zeta^l)) \leq \sigma^2, \tag{2}$$

$$\mathbb{E}H^l(x, \zeta^l) = \nabla F^l(x) - \nabla F^{l-1}(x), \mathbb{V}(H^l(x, \zeta^l)) \leq V_l := M_b 2^{-bl}. \tag{3}$$

(c). *The cost to query $\mathcal{SO}^l$ is bounded: $C_l \leq M_c 2^{cl}$.*

(d). *The objective $F$ and the approximation function $F^l$ are $S_F$-smooth.*

Assumptions 2.1 (a) and (c) imply that obtaining unbiased gradient estimator of more accurate approximation functions $F^l$ (namely smaller bias from the true gradient) requires higher cost. The variance decay of the difference estimator $H^l(x, \zeta^l)$ in Assumption 2.1(b) is the key assumption for MLMC methods. Intuitively, since $\nabla F^l(x) - \nabla F^{l-1}(x)$ becomes very small for large $l$, constructing highly correlated estimators of $\nabla F^l(x)$ and $\nabla F^{l-1}(x)$ using the same samples $\zeta^l$ will likely yield small variance of their difference.

In the nonconvex setting, since we care about $\epsilon$-stationarity point rather than function values, we use the following assumption on approximation error of the gradient.

**Assumption 2.2.** *There exists $M_a > 0$ such that for any $l \in \mathbb{N}$, it holds*

$$\|\nabla F^l(x) - \nabla F(x)\|_2^2 \leq B_l := M_a 2^{-al}.$$

*Remark* 2.1 (**Relationship between the assumptions**). Our main results in the strongly convex case can be easily extended to the PL condition [24]. Particularly, under PL condition, Assumption 2.1(a) and Assumption 2.2 are exchangeable for biased gradient-based methods to achieve the same convergence and total cost results. We defer the related discussion to Appendix D.1. In general, these two assumptions do not imply each other. Both of them hold for convex CSO problems, as we will show in Section 4. In general, these two assumptions reflect two different ways of constructing approximations. Constructing uniform function approximation first then taking gradient is more suitable for a problem when the form of the true gradient is unknown, for instance, DRO problems. On the other hand, directly constructing gradient approximations is more suitable when one knows the form of the true gradient and particularly when the approximated gradient does not have a corresponding objective function.

## 2.2 SGD with MLMC Estimators

In the sequel, we construct a direct biased gradient estimator and four different multilevel Monte Carlo gradient estimators based on the oracle $\mathcal{SO}^l$. These estimators will be fed in the generic stochastic gradient descent (SGD) algorithm given as follows.

---
**Algorithm 1** SGD Framework

---
**Input:** Number of iterations $T$, stepsizes $\{\gamma_t\}_{t=1}^T$, initialization point $x_1$.
1: **for** $t = 1$ to $T$ **do**
2:     Construct a gradient estimator $v(x_t)$ of $\nabla F(x_t)$.
3:     Update $x_{t+1} = x_t - \gamma_t v(x_t)$.
4: **end for**
**Output:** $\{x_t\}_{t=1}^T$.

---

Due to potential computing limitation, one might only have access to the stochastic oracles of the first $L$ levels of approximation functions $\{F^l(x)\}_{l=0}^L$, for some $L > 0$. Let $\{n_l\}_{l=0}^L$ denote some batch sizes.

**$L$-SGD estimator:** at a query point $x$, query oracle $\mathcal{SO}^L$ at point $x$ for $n_L$ times to obtain $\{h^L(x, \zeta_i^L)\}_{i=1}^{n_L}$, and then construct

$$v^{L\text{-SGD}}(x) := \tfrac{1}{n_L} \sum_{i=1}^{n_L} h^L(x, \zeta_i^L). \tag{4}$$

**V-MLMC estimator:** at a query point $x$, query the oracle $\mathcal{SO}^l$ at $x$ for $n_l$ times to obtain $\{H^l(x, \zeta_i^l)\}_{i=1}^{n_l}$ for $l = 0, ..., L$, and then construct

$$v^{\text{V-MLMC}}(x) := \sum_{l=0}^L \tfrac{1}{n_l} \sum_{i=1}^{n_l} H^l(x, \zeta_i^l). \tag{5}$$

**RT-MLMC estimator**: at a query point $x$, first sample a random level $\iota$ according to the probability distribution $Q_1 = \{q_l\}_{l=0}^L$, such that $P(\iota = l) = q_l$, where $\sum_{l=0}^L q_l = 1$, then query only oracle $\mathcal{SO}^\iota$ at $x$ once to obtains $H^\iota(x, \zeta^\iota)$.

$$v^{\text{RT-MLMC}}(x) := q_\iota^{-1} H^\iota(x, \zeta^\iota). \tag{6}$$

**RU-MLMC estimator**: at a query point $x$, first sample a random level $\iota$ according to the probability distribution $Q_2 = \{q_l\}_{l=0}^\infty$, such that $P(\iota = l) = q_l$, where $\sum_{l=0}^\infty q_l = 1$, then query only oracle $\mathcal{SO}^\iota$ at $x$ once to obtains $H^\iota(x, \zeta^\iota)$.

$$v^{\text{RU-MLMC}}(x) := q_\iota^{-1} H^\iota(x, \zeta^\iota). \tag{7}$$

**RR-MLMC estimator**: at a query point $x$, first sample a random level $L$ according to the probability distribution $Q_2 = \{q_l\}_{l=0}^\infty$, such that $P(L = l) = q_l$, where $\sum_{l=0}^\infty q_l = 1$, then query oracle $\mathcal{SO}^l$ at point $x$ once for $l = 0, .., L$ to obtain $\{H^l(x, \zeta^l)\}_{l=0}^L$.

$$v^{\text{RR-MLMC}}(x) := \sum_{l=0}^L p_l H^l(x, \zeta^l), \tag{8}$$

where $p_l = \frac{1}{1 - \sum_{l'=0}^{l-1} q_{l'}}$, and $\sum_{l'=0}^{-1} q_{l'} = 0$.

Note that the V-MLMC estimator requires querying the stochastic oracles for each level $l = 0, \ldots, L$ with some fixed $L$. The other three MLMC estimators only query the stochastic oracle from a random level or a random subset of levels, with appropriate reweighing. As an immediate observation, we summarize the bias, variance, and (expected) computation cost in Table 2 when $x \in \mathbb{R}^d$ is independent from $v^A$ for $A$ being $L$-SGD, V-MLMC, RT-MLMC, RU-MLMC, RR-MLMC. In the table, $C_{\text{iter}}^A$ denotes the expected computation cost for constructing estimator $A$. See Lemmas B.1 and B.2 for a detailed derivation.

Both RU-MLMC and RR-MLMC give unbiased gradient estimators of $F$, whereas the other three estimators are biased. As a result, combining SGD with $L$-SGD estimator, V-MLMC estimator, and RT-MLMC estimator will only lead to a stationary point of the approximation function $F^L(x)$, assuming the variance of these estimators are bounded.

Table 2: Bias, Variance, and Cost of Gradient Estimators.

| Estimators $A$ | Expectation $\mathbb{E}v^A(x)$ | Variance $\mathbb{V}(v^A(x))$ | Cost $C^A_{\text{iter}}$ |
|---|---|---|---|
| $L$-SGD | $\nabla F^L(x)$ | $n_L^{-1}\sigma^2$ | $n_L C_L$ |
| V-MLMC | $\nabla F^L(x)$ | $\sum_{l=0}^{L} n_l^{-1} V_l$ | $\sum_{l=0}^{L} n_l C_l$ |
| RT-MLMC | $\nabla F^L(x)$ | $\sum_{l=0}^{L} q_l^{-1} V_l$ | $\sum_{l=0}^{L} q_l C_l$ |
| RU-MLMLC | $\nabla F(x)$ | $\sum_{l=0}^{\infty} q_l^{-1} V_l$ | $\sum_{l=0}^{\infty} q_l C_l$ |
| RR-MLMC | $\nabla F(x)$ | $\sum_{L=0}^{\infty} q_L \left( \sum_{l=0}^{L} p_l^2 V_l \right)$ | $\sum_{L=0}^{\infty} q_L \left( \sum_{l=0}^{L} C_l \right)$ |

*Remark* 2.2 (**Conditions and computation costs of RU-MLMC and RR-MLMC**). For both RU-MLMC and RR-MLMC, invoking the definitions of $C_l$ and $V_l$, it can be easily seen that when $b \leq c$, either the variance or the expected per-iteration cost will be unbounded. Hence, these two unbiased MLMC estimators are only useful in the restrictive regime when $b > c$. In fact, when $b > c$, by setting $q_l \propto 2^{-(b+c)l/2}$, we immediately have that $\mathbb{V}(v^A(x)) = \mathcal{O}(1), C^A_{\text{iter}} = \mathcal{O}(1)$, for A being either RU-MLMC or RR-MLMC. As a result, the convergence rates of SGD based on RU-MLMC and RR-MLMC estimators can be directly obtained from the standard results for unbiased SGD; see, e.g., [6, 29, 13]. In particular, the iteration complexities (hence, expected total costs) of RU-MLMC or RR-MLMC to achieve an $\epsilon$-optimal solution are $\mathcal{O}(1/\epsilon)$ for strongly convex $F$ and $\mathcal{O}(1/\epsilon^2)$ for general convex $F$, under appropriately chosen stepsize. For nonconvex smooth objective $F$, the expected total cost for achieving an $\epsilon$-stationary point such that $\mathbb{E}[\|\nabla F(x)\|_2^2] \leq \epsilon$ is $\mathcal{O}(1/\epsilon^4)$. For completeness, we provide detailed analysis in Appendix B about the bias, the variance, and the per-iteration cost and in Appendix C.4 about the total cost.

For ease of notation, let $\widehat{x}_T^A$ denote the output of SGD using estimator $A$ after $T$ iterations, where $A$ corresponds to $L$-SGD, V-MLMC, or RT-MLMC. Under Assumption 2.2 we have the decomposition of errors in the (strongly) convex case:

$$\mathbb{E}[\underbrace{F(\widehat{x}_T^A) - F(x^*)}_{\text{Error of Algorithm } A \text{ on } F}] = \mathbb{E}[\underbrace{F(\widehat{x}_T^A) - F^L(\widehat{x}_T^A)}_{\text{Approximation Error} \leq B_L} + \underbrace{F^L(\widehat{x}_T^A) - F^L(x^L)}_{\text{Error of SGD on } F^L}]$$
$$+ \underbrace{F^L(x^L) - F^L(x^*)}_{\leq 0 \text{ by optimality of } x^L} + \underbrace{F^L(x^*) - F(x^*)}_{\text{Approximation Error} \leq B_L}, \quad (9)$$

where $x^L$ is the minimizer of $F^L(x)$. The decomposition suggests that as long as the level $L$ is large enough, e.g., $L = \lceil a^{-1} \log(4M_a\epsilon^{-1}) \rceil$ such that $2B_L \leq \epsilon/2$, and the number of iteration $T$ is large enough such that expected error of SGD on $F^L$ is smaller than $\epsilon/2$, the output of these methods will be $\epsilon$-optimal in the convex setting.

*Remark* 2.3 (**Computation costs of $L$-SGD**). For $L$-SGD, the total cost is given by $T \cdot (n_L C_L)$. With the above choice of $L$, we have $C_L = \mathcal{O}(\epsilon^{-c/a})$. From standard analysis of SGD (see Theorem A.1 in Appendix A), if the variance of the gradient estimator is $\mathcal{O}(1)$, we have $T = \mathcal{O}(\frac{\sigma^2}{n_L \epsilon})$ when $F^L$ is strongly convex [6], and we have $T = \mathcal{O}(\frac{\sigma^2}{n_L \epsilon^2})$ when $F^L$ is convex and smooth. See Appendix C.1 for detailed results on $L$-SGD. Therefore, the total computation costs for $L$-SGD to achieve an $\epsilon$-optimal solution are $O(\epsilon^{-1-c/a})$ and $O(\epsilon^{-2-c/a})$, respectively in the (strongly) convex settings.

## 3 Total Cost Analysis of V-MLMC and RT-MLMC

This section mainly focuses on the total cost analysis of the two biased MLMC gradient methods, V-MLMC and RT-MLMC. Due to the page limit, we only demonstrate the regime when $b \geq c$, i.e., the decrease rate of the variance is larger than or equal to the increase rate of the cost. Full results can be found in Appendix C. Let $\widehat{x}_T^A$ be selected uniformly from $x_1^A, ..., x_T^A$ for $A$ being V-MLMC or RT-MLMC.

**Theorem 3.1** (Expected Total cost of RT-MLMC). *For RT-MLMC with a distribution $Q = \{q_l\}_{l=0}^L$ such that $q_l \propto 2^{-(b+c)l/2}$, when $b \geq c$, its expected total cost satisfies the followings.*

- *If Assumption 2.1 holds and $F^L$ is strongly convex, let $\gamma_t = 1/(t + S_F^2/\mu^2)$ and $L = \lceil 1/a \log(4M_a\epsilon^{-1}) \rceil$. Then $x_T^{RT\text{-}MLMC}$ is an $\epsilon$-optimal point of $F$ after $T = \mathcal{O}(\epsilon^{-1})$ iterations. The total cost of RT-MLMC satisfies $C = \widetilde{\mathcal{O}}(\epsilon^{-1})$.*

- *If Assumption 2.1 holds and $F^L$ is convex, let stepsizes $\gamma_t = \mathcal{O}(1/\sqrt{T})$ and $L = \lceil 1/a \log(4M_a \epsilon^{-1}) \rceil$. Then $\widehat{x}_T^{RT\text{-}MLMC}$ is an $\epsilon$-optimal point of $F$ after $T = \mathcal{O}(\epsilon^{-2})$ iterations, and the total cost of RT-MLMC satisfies $C = \widetilde{\mathcal{O}}(\epsilon^{-2})$.*

- *If Assumption 2.1(b)(c)(d) and 2.2 hold, let stepsizes $\gamma_t = \mathcal{O}(1/\sqrt{T})$ and $L = \lceil 1/a \log(4M_a \epsilon^{-2}) \rceil$. Then $\widehat{x}_T^{RT\text{-}MLMC}$ is an $\epsilon$-stationarity point of $F$ after $T = \mathcal{O}(\epsilon^{-4})$ iterations. The total cost of RT-MLMC satisfies $C = \widetilde{\mathcal{O}}(\epsilon^{-4})$.*

*Remark* 3.1 (Selection of $\{q_l\}_{l=0}^L$). We use the RT-MLMC for convex objectives to illustrate how we select the $q_l$. To minimize the total cost for achieving $\epsilon$-optimality, conceptually, we are solving the following optimization problem:

$$\min_{\{q_l\}_{l=0}^L} \quad T^{\mathrm{RT-MLMC}} C_{\mathrm{iter}}^{\mathrm{RT-MLMC}}$$

$$\text{s.t.} \quad (a) \sum_{l=0}^L q_l = 1, \quad (b) \, q_l \geq 0, \forall\, l = 0, ..., L. \tag{10}$$

where the constraints are to make sure that $Q = \{q_l\}_{l=0}^L$ is a distribution. Problem (10) is nonconvex and we cannot directly solve it. Later in Section C, we show that under properly selected stepsizes, it holds that

$$T^{\mathrm{RT-MLMC}} C_{\mathrm{iter}}^{\mathrm{RT-MLMC}} \propto \mathbb{V}(v^{\mathrm{RT-MLMC}}) C_{\mathrm{iter}}^{\mathrm{RT-MLMC}} = \Big(\sum_{l=0}^L q_l C_l\Big)\Big(\sum_{l=0}^L V_l/q_l\Big).$$

Optimizing the Lagrange function over $\{q_l\}_{l=0}^L$ gives $q_l \propto 2^{-(b+c)l/2}$. Although they might not be the optimal solution of (10), we have shown that such choice of $q_l$ can greatly reduce the total cost significantly comparing to L-SGD.

Note that when $b > c$, the total cost of RT-MLMC gets rid of the dependence on $\log(\epsilon^{-1})$. Similar to RU-MLMC and RR-MLMC, RT-MLMC achieves an $\mathcal{O}(1)$ expected per-iteration cost and $\mathcal{O}(1)$ variance.

**Theorem 3.2** (Total cost for V-MLMC). *When $b \geq c$, for V-MLMC with batch sizes $n_l = \lceil 2^{-(b+c)l/2} N \rceil$ for some $N > 0$, its total cost satisfies the followings.*

- *If Assumption 2.1 holds and $F^L$ is strongly convex, let stepsizes $\gamma_t = 1/S_F$, $L = \lceil 1/a \log(4M_a \epsilon^{-1}) \rceil$, and $N = \widetilde{\mathcal{O}}(\epsilon^{-1})$. Then $x_T^{V\text{-}MLMC}$ is an $\epsilon$-optimal solution of $F$ after $T = \mathcal{O}(\log(\epsilon^{-1}))$ iterations. The total cost of V-MLMC satisfies $C = \widetilde{\mathcal{O}}(\epsilon^{-\max\{1,c/a\}})$.*

- *If Assumption 2.1 holds and $F^L$ is convex, let stepsizes $\gamma_t = 1/(2S_F)$, $L = \lceil 1/a \log(4M_a \epsilon^{-1}) \rceil$, and $N = \widetilde{\mathcal{O}}(\epsilon^{-1})$. Then $\widehat{x}_T^{V\text{-}MLMC}$ is an $\epsilon$-optimal solution of $F$ after $T = \mathcal{O}(\epsilon^{-1})$ iterations. The total cost of V-MLMC satisfies $C = \widetilde{\mathcal{O}}(\epsilon^{-1-\max\{1,c/a\}})$.*

- *If Assumption 2.1(b)(c)(d) and 2.2 hold, let stepsizes $\gamma_t = 1/(S_F)$, $L = \lceil 1/a \log(4M_a \epsilon^{-2}) \rceil$, and $N = \widetilde{\mathcal{O}}(\epsilon^{-2})$. Then $\widehat{x}_T^{V\text{-}MLMC}$ is an $\epsilon$-stationarity point of $F$ after $T = \mathcal{O}(\epsilon^{-2})$ iterations. The total cost of V-MLMC satisfies $C = \widetilde{\mathcal{O}}(\epsilon^{-2-2\max\{1,c/a\}})$.*

*Remark* 3.2. Optimizing the total cost with respect to $\{n_l\}_{l=0}^L$ in the continuous space shows that $n_l \propto 2^{-(b+c)l/2}$. Since mini-batch size $n_l$ has to be integer numbers, we set $n_l = \lceil 2^{-(b+c)l/2} N \rceil$ for some constant $N > 0$. Since $n_l \leq 2^{-(b+c)l/2} N + 1$, the per-iteration cost satisfies

$$C_{\mathrm{iter}}^{\mathrm{V-MLMC}} \leq \sum_{l=0}^L 2^{(c-b)l/2} N + \sum_{l=0}^L 2^{cl} \stackrel{(\text{if } b>c)}{=} \mathcal{O}(N) + \mathcal{O}(\epsilon^{-c/a}). \tag{11}$$

where $L = \lceil 1/a \log(4M_a \epsilon^{-1}) \rceil$ as specified in the theorem. The first term in (11) represents the desired balanced per-iteration cost of MLMC, while the second term refers to the cost incurred by rounding to integer numbers. When $c < b$ and $N = \mathcal{O}(1)$, under the specified $L$ in the theorem, the second term dominates the per-iteration cost. At the same time, since the variance of V-MLMC is $\mathcal{O}(N^{-1}) = \mathcal{O}(1)$, the iteration complexity and the per-iteration cost of V-MLMC are the same as $L$-SGD. Therefore V-MLMC cannot reduce the total cost comparing to $L$-SGD when $N$ is small.

Using large $N = \mathcal{O}(\epsilon^{-1})$ reduces the variance to $\mathcal{O}(\epsilon)$ and V-MLMC behaves like gradient descent. Thus the iteration complexity of V-MLMC is reduced by $\mathcal{O}(\epsilon^{-1})$ in the strongly convex case and the convex case and the total cost becomes $\widetilde{\mathcal{O}}(\epsilon^{-\max\{1,c/a\}})$ and $\mathcal{O}(\epsilon^{-1-\max\{1,c/a\}})$, respectively. In any case, V-MLMC is always $\mathcal{O}(\epsilon^{-\min\{1,c/a\}})$ better than $L$-SGD.

One may also use $N = k2^{(b+c)L/2}$ with certain $k$ so that $2^{-(b+c)l/2}N$ is integer for any $l \in [L]$. Then there will be no extra cost incurred by rounding to integers, i.e., the second term on the right hand side of (11) becomes 0. In such cases, the total cost of V-MLMC matches that of RT-MLMC and is always $\mathcal{O}(\epsilon^{-c/a})$ better than $L$-SGD.

The drawback is that V-MLMC has to use very large $N$ and thus large mini-batch $\{n_l\}_{l=0}^L$ for small $l$. Note that $L$-SGD cannot use large batch to reduce the total cost.

When $b > c$, the total cost of V-MLMC can get rid of the dependence on $\log(\epsilon^{-1})$ in the convex and nonconvex smooth case by setting $N = \mathcal{O}(\epsilon^{-1})$ or $N = \mathcal{O}(\epsilon^{-2})$ in the convex case or the nonconvex case, respectively. The logarithmic term in the strongly convex case comes from the iteration complexity. When $b = c$, $\mathcal{O}(N)$ in (11) becomes $\mathcal{O}(N\log(\epsilon^{-1}))$ and when $b < c$, $\mathcal{O}(N)$ in (11) becomes $\mathcal{O}(N\epsilon^{-(c-b)/2a})$. Note that the variance will increase by the same amount, which explains why the cost of the MLMC methods increases gets larger when $b \le c$. Detailed discussions are in Appendix C.3.

*Remark* 3.3 (**Applicability of MLMC gradient methods**). When $b > c$, RU-MLMC and RR-MLMC are the most favorable among the four MLMC methods since they have unbiased gradient estimators and do not need to specify $L$ in advance. When $b \le c$, only RT-MLMC and V-MLMC are applicable. They introduce bias to avoid the high computation cost. RT-MLMC is the most versatile algorithm among the four MLMC methods since it has no restrictions on $a, b, c$ and does not need any mini-batch. It suits the situations when the per-iteration budget is limited, e.g., when one only has limited samples or computation power. V-MLMC uses large mini-batches that lead to a very small variance. Therefore a constant stepsize $\mathcal{O}(1)$ is sufficient to guarantee convergence.

*Remark* 3.4 (**Unbiased gradient estimator is not necessarily the best**). When $b \le c$, RU-MLMC and RR-MLMC can still build up unbiased gradient estimators. However, the variance and the expected cost cannot stay finite at the same time. As a result, the expected total cost for achieving $\epsilon$-optimality becomes infinite. Such observation suggests that unbiased gradient estimator are not necessarily the best. It further strengthens the importance of balancing the bias, variance, and cost.

## 4   Applications to Conditional Stochastic Optimization

Conditional stochastic optimization has been utilized to model and solve many applications in machine learning, including the optimal control in linearly-solvable Markov decision process [8], policy evaluation and control in reinforcement learning [8, 9, 28], meta-learning [20], instrumental variable regression [27]. Previously, Hu et al. [20] considered biased SGD method and biased variance reduction methods using SPIDER [12]. Note that the biased SGD method they used can be treated as a special case of the $L$-SGD in our paper.

In the following, we show that MLMC methods can significantly reduce the sample complexity and achieve better results than the biased variance reduction methods proposed in their paper. Recall the definition of CSO problem and its approximation function $F^l(x)$.

$$\min_{x \in \mathbb{R}^d} F(x) := \mathbb{E}_\xi f_\xi(\mathbb{E}_{\eta|\xi} g_\eta(x, \xi)); \; F^l(x) = \mathbb{E}_{\xi^l} \mathbb{E}_{\{\eta_j^l\}_{j=1}^{2^l}|\xi^l} \left[ f_{\xi^l} \left( \frac{1}{2^l} \sum_{j=1}^{2^l} g_{\eta_j^l}(x, \xi^l) \right) \right], \quad (12)$$

where $\xi^l \sim \mathbb{P}(\xi)$ and $\{\eta_j^l\}_{j=1}^{2^l} \sim \mathbb{P}(\eta|\xi_i^l)$. Denote $\zeta^l = (\xi^l, \{\eta_j^l\}_{j=1}^{2^l})$ and $\widehat{g}_{n_1:n_2}(x, \zeta^l) = (n_2 - n_1 + 1)^{-1} \sum_{j=n_1}^{n_2} g_{\eta_j^l}(x, \xi^l)$ for some $1 \le n_1 \le n_2$. For each query on a query point $x$, $\mathcal{SO}^l$ returns $(h^l(x, \zeta^l), H^l(x, \zeta^l))$ such that

$$h^l(x, \zeta^l) = \nabla \widehat{g}_{1:2^l}(x, \zeta^l)^\top \nabla f_{\xi^l}(\widehat{g}_{1:2^l}(x, \zeta^l));$$
$$H^l(x, \zeta^l) = \nabla_x \left[ f_{\xi^l}(\widehat{g}_{1:2^l}(x, \zeta^l)) - \frac{1}{2} f_{\xi^l}(\widehat{g}_{1:2^{l-1}}(x, \zeta^l)) - \frac{1}{2} f_{\xi^l}(\widehat{g}_{1+2^{l-1}:2^l}(x, \zeta^l)) \right]. \quad (13)$$

Table 3: Comparison of Algorithms on CSO

| Convexity of $F$ | V-MLMC | RT-MLMC | RU-MLMC/RR-MLMC | [20] |
|---|---|---|---|---|
| Strongly Convex | $\widetilde{\mathcal{O}}(\epsilon^{-1})$ | $\widetilde{\mathcal{O}}(\epsilon^{-1})$ | N.A. | $\widetilde{\mathcal{O}}(\epsilon^{-2})$ |
| Convex | $\widetilde{\mathcal{O}}(\epsilon^{-2})$ | $\widetilde{\mathcal{O}}(\epsilon^{-2})$ | N.A. | $\mathcal{O}(\epsilon^{-3})$ |
| Nonconvex Smooth | $\widetilde{\mathcal{O}}(\epsilon^{-4})$ | $\widetilde{\mathcal{O}}(\epsilon^{-4})$ | N.A. | $\mathcal{O}(\epsilon^{-6})^{\dagger}$ |
| $\dagger$ becomes $\mathcal{O}(\epsilon^{-5})$ when further applying variance reduction. | | | | |

We denote the cost as the total number of samples $\eta$. Thus the cost to query oracle $\mathcal{SO}^l$ is $C_l = 2^l$. To apply the MLMC methods, it remains to verify that Assumption 2.1 and Assumption 2.2 hold. We follow Hu et al. [20] and make some assumptions on the CSO problem.

**Assumption 4.1.** *We assume that the followings hold for CSO problems.*

- $\sigma_g^2 := \sup_{x \in \mathbb{R}^d, \xi} \mathbb{E}_{\eta|\xi} ||g_\eta(x, \xi) - \mathbb{E}_{\eta|\xi} g_\eta(x, \xi)||_2^2 < +\infty.$

- $f_\xi(\cdot)$ *is $S_f$-smooth and $L_f$-Lipschitz continuous;*

- $g_\eta(\cdot, \xi)$ *is $S_g$-smooth and $L_g$-Lipschitz continuous.*

**Proposition 4.1.** *Under Assumption 4.1, we have the following results:*

- *The functions $F$ and $F^l$ are $(S_g L_f + S_f L_g^2)$-smooth for any $l \in \mathbb{N}$. It holds that*

$$\|\nabla F^l(x) - \nabla F(x)\|_2^2 \leq L_g^2 S_f^2 \sigma_g^2 2^{-l} \text{ and } |F^l(x) - F(x)| \leq S_f \sigma_g^2 2^{-l}.$$

- *The variance of the oracle $\mathcal{SO}^l$ satisfies*

$$\text{Var}(h^l(x, \zeta^l)) \leq L_f^2 L_g^2 \text{ and } \text{Var}(H^l(x, \zeta^l)) \leq L_g^2 S_f^2 \sigma_g^2 2^{-l}.$$

Proposition 4.1 implies that under Assumption 4.1, CSO problems have the following parameter:

$$a = 1, b = 1, c = 1.$$

Since $b = c$, RU-MLMC and RR-MLMC are not applicable. By Theorem 3.2 and Theorem 3.1, we have the total sample complexity of RT-MLMC and V-MLMC, which is summarized in Table 3.

**Corollary 4.1.** *Under Assumption 4.1, for strongly convex CSO problem, the sample complexity of V-MLMC and RT-MLMC for finding $\epsilon$-optimal solution is $\widetilde{\mathcal{O}}(\epsilon^{-1})$; for convex CSO problem, the sample complexity of V-MLMC and RT-MLMC is $\widetilde{\mathcal{O}}(\epsilon^{-2})$; for nonconvex smooth CSO problem, the sample complexity of V-MLMC and RT-MLMC for finding $\epsilon$-stationary point is $\widetilde{\mathcal{O}}(\epsilon^{-4})$.*

*Remark* 4.1. The sample complexities of V-MLMC and RT-MLMC are better than the lower bounds proved in Hu et al. [20]. The reason is that Hu et al. [20] assumed access to a stochastic gradient oracle (black-box oracle) that returns a biased gradient of CSO with $\mathcal{O}(\epsilon)$ bias and variance $\mathcal{O}(1)$ at a query cost of $\mathcal{O}(\epsilon^{-1})$. In contrast, our paper shows that using RT-MLMC, one can construct a specific gradient estimator (white-box oracle) that has the same $\mathcal{O}(\epsilon)$ bias and $\mathcal{O}(1)$ variance, but the cost to query such oracle is of order $\mathcal{O}(\log(\epsilon^{-1}))$. Since the iteration complexity (which only depends on the bias and the variance) stays the same, the reduced cost to query the oracle leads to a reduced total cost. Such observation further strengthens the importance of taking into account the oracle cost when studying the complexity of biased oracle models.

## 5 Numerical Experiments

In numerical experiments, we apply four MLMC gradient methods and LSGD on three problems, a synthetic problem with biased oracles, invariant least square, and invariant absolute regression.

The synthetic problem with biased oracles is of the form: $\min_{x \in \mathbb{R}^d} F(x) := \frac{1}{2}\|x - z_\infty\|^2$, where $z_\infty = \lim_{n \to \infty} z_n$ and stochastic observation of $z_n$ can be obtained via some simulation process with cost $n$. Thus the approximation function is $F^l(x) = \frac{1}{2}\|x - z_{2^l}\|^2$. Let $z_n = (1 + \text{bias}/n)\mathbf{1}_d$ where $\widehat{z}_n$ is the output of a simulation such that $\widehat{z}_n \sim \mathbf{N}(z_n, \sigma^2 \mathbb{I}_d)$. Here bias is a hyperparameter that controls the bias.

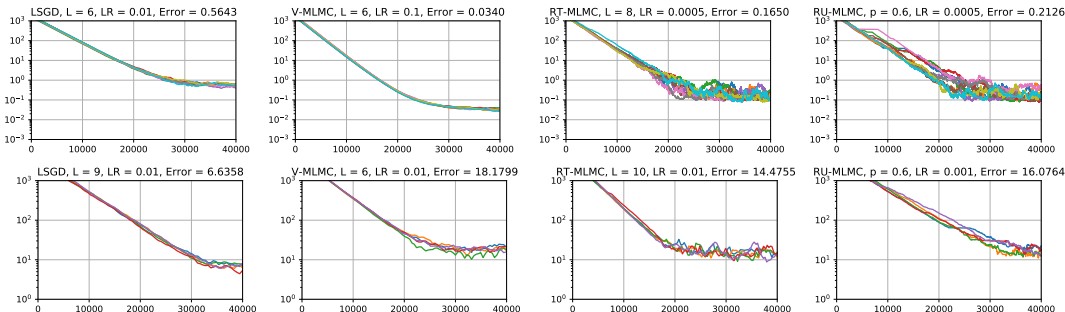

Figure 1: Top row: synthetic problem. Bottom row: invariant least square. "LR": learning rate or stepsizes. "Error": average error of last iterate. Each subfigure represents the best average last iterate error a method can achieve with truncation level $L$ selected within $\{0, ..., 10\}$, geometry distribution with parameter $p$ within $\{0.1, ..., 0.9\}$, and stepsizes. Each line in the subfigure represents an instance of the method with a different random initialization. For the synthetic problem, we do not use any mini-batches, while for invariant least square, we apply mini-batches on randomized MLMC methods to control variance. The performance is measured by the average error of last iterate.

The invariant least square is a special case of CSO problem with linear function and is of the form:

$$\min_{x \in \mathbb{R}^d} \mathbb{E}_{\xi=(a,b)} f(\mathbb{E}_{\eta|a}\eta^\top w - b),$$

where $f$ is a loss function, $\xi = (a, b)$ represents the feature label pair, $\eta$ represents a perturbed feature. For $f$, we consider absolute loss and square loss. For simplicity, let $\eta \sim \mathbf{N}(a, \sigma^2 \mathbb{I}_d)$. We generate 2000 sample from $\xi$, i.e., 2000 feature label pairs such that $a$ follows some multivariate normal distribution and $b = a^\top x^*$ with $x^*$ generated via a multivariate normal distribution. The MLMC gradient is constructed following (13).

Figure 1 summarizes the optimal parameter setup that achieves the smallest average error over a certain number of trials under a given total budget for quadratic program and invariant least square. More detailed setup and report on comparison and discussion on RR-MLMC and invariant absolute regression are given in Appendix F.

- We observe that V-MLMC has the smallest variance as suggested by the theory while the variances of RT-MLMC, RU-MLMC, and RR-MLMC are larger than the variance of LSGD due to the extra randomness caused by additional sampling. Thus we use mini-batch for randomized MLMC methods to obtain a stable training process while using larger stepsizes.

- In practice, we observe that sometimes LSGD outperforms MLMC gradient methods, especially when we are only aiming for low accuracy solution. This may be caused by the high variance of MLMC methods and possibly large hidden constant factor in the complexity bounds. When using the same truncation level, we observe that RT-MLMC generally converges faster than LSGD in the early stages but suffers more from the high variance in later stages.

- We observe that biased MLMC methods generally outperform unbiased MLMC methods. It further justifies the importance of bias, variance, cost tradeoff as we mentioned in Remark 3.4.

## 6  Conclusion

This paper provides a systematic study of the bias-variance-cost tradeoff of several MLMC gradient methods under a generic biased oracle model for stochastic optimization, shedding light on their superiority and limitations under different situations. For future work, combing variance reduction techniques with MLMC gradient estimators should further reduce the total costs in the nonconvex smooth setting. Second, the current MLMC techniques focus on constructing gradient estimators whose bias level maintains the same at each iteration. Another possible approach is to construct MLMC gradient estimators that adaptively reduce the bias. Lastly, it remains interesting to explore the power of MLMC gradient methods on other applications, for instance, quantization in distributed optimization and federated learning. It also remains open to characterize the conditions when MLMC can greatly outperform LSGD numerically in real-world applications.

## Acknowledgments and Disclosure of Funding

We thank all reviewers and the area chair for the suggestions to the paper. We thank the reviewer of our last NeurIPS submission [20] for pointing out the potential of multilevel Monte Carlo methods on conditional stochastic optimization. This work is supported by NSF CRII 1755829.

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
