# Appendix

## A Preliminaries on Stochastic Gradient Descent

The following lemma demonstrates the convergence property of the SGD framework when the gradient estimator $v(x)$ is unbiased and has bounded variance. Similar results can be found in Nemirovski et al. [29], Rakhlin et al. [31]. Differently, these papers assumed that the norm of the gradient estimator, i.e., $\|v(x)\|_2$, is bounded, whereas we only assume that the variance of the gradient estimator is bounded. One can also refer to the recent summary [6] for more general results on SGD.

Note that $L$-SGD, V-MLMC, and RT-MLMC are biased methods for optimizing $F$ but unbiased methods for optimizing $F^L$. We use Lemma A.1 together with Assumption 2.1(a) and equation (9) to derive the global convergence of biased methods on $F$ in the (strongly) convex setting; with Assumption 2.2 and equation (21) to derive the stationary convergence of biased methods on $F$ in the nonconvex smooth setting.

**Lemma A.1** (Convergence of SGD). *Suppose that $F(x)$ is $S_F$-smooth on $\mathbb{R}^d$ and suppose that there exists a constant $V > 0$ such that*

$$\mathbb{E}v(x) = \nabla F(x), \mathbb{V}(v(x)) \leq V,$$

*where expectations are taken w.r.t the randomness in $v$. Let $x^*$ be a minimizer of $F(x)$ on $\mathbb{R}^d$ and $\widehat{x}_T$ be selected uniformly randomly from $\{x_t\}_{t=1}^T$. We have the following results:*

*(a1). If $F(x)$ is $\mu$-strongly convex, for fixed stepsizes $\gamma_t = \gamma \in (0, \frac{1}{S_F}]$, we have*

$$\mathbb{E}[F(x_T) - F(x^*)] \leq (1 - \gamma\mu)^{T-1}[F(x_1) - F(x^*)] + \frac{S_F \gamma V}{2\mu}.$$

*(a2). If $F(x)$ is $\mu$-strongly convex, for stepsizes $\gamma_t = \frac{1}{\mu(t + 2S_F^2/\mu^2)}$, we have*

$$\mathbb{E}[F(x_T) - F(x^*)] \leq \frac{S_F \max\{V, \mu^2(1 + 2S_F^2/\mu^2)\|x_1 - x^*\|_2^2\}}{\mu^2(T + 2S_F^2/\mu^2)}.$$

*(b). If $F(x)$ is convex, for stepsizes $\gamma_t = \gamma \in (0, \frac{1}{2S_F}]$, we have*

$$\mathbb{E}[F(\widehat{x}_T) - F(x^*)] \leq \gamma V + \frac{\|x_1 - x^*\|_2^2}{\gamma T}.$$

*(c). If $F(x)$ is nonconvex, for fixed stepsizes $\gamma_t = \gamma \in (0, \frac{1}{S_F}]$, we have*

$$\mathbb{E}\|\nabla F(\widehat{x}_T)\|_2^2 \leq \frac{2(F(x_1) - F(x^*))}{\gamma T} + S_F \gamma V.$$

Note that we do not specify the stepsizes in cases (a1), (b), and (c). When the variance of $v(x)$ is of order $\mathcal{O}(\epsilon)$, one can use stepsizes that are independent of $\epsilon$ to guarantee $\epsilon$-optimality or $\epsilon$-stationarity. The algorithm would behave similar like gradient descent. On the other hand, when the variance of $v(x)$ is of order $\mathcal{O}(1)$ or even larger, one should use stepsizes $\gamma_t = \mathcal{O}(1/t)$ in the strongly convex setting and $\gamma_t = \mathcal{O}(1/\sqrt{T})$ in the convex or nonconvex smooth setting.

*Proof.* Notice that $x^*$ is a minimizer of $F$ over $\mathbb{R}^d$. We have $\nabla F(x^*) = 0$.

**Strongly Convex Case (a1)** By smoothness, one has that

$$F(x_{t+1}) - F(x_t) \leq \nabla F(x_t)^\top (x_{t+1} - x_t) + \frac{S_F}{2}\|x_{t+1} - x_t\|_2^2$$

$$= -\gamma_t \nabla F(x_t)^\top v(x_t) + \frac{S_F \gamma_t^2}{2}\|v(x_t)\|_2^2.$$

Taking expectation conditioned on $x_t$ on both side, since $\mathbb{E}[v(x_t)|x_t] = \nabla F(x_t)$ it holds that

$$
\begin{aligned}
\mathbb{E}[F(x_{t+1}) - F(x_t)|x_t] \leq & -\gamma_t\|\nabla F(x_t)\|_2^2 + \frac{S_F\gamma_t^2}{2}\mathbb{E}[\|v(x_t)\|_2^2|x_t] \\
= & -\gamma_t\|\nabla F(x_t)\|_2^2 + \frac{S_F\gamma_t^2}{2}\|\nabla F(x_t)\|_2^2 + \frac{S_F\gamma_t^2}{2}\mathbb{V}(\|v(x_t)\|_2^2|x_t) \\
= & -\left(\gamma_t - \frac{S_F\gamma_t^2}{2}\right)\|\nabla F(x_t)\|_2^2 + \frac{S_F\gamma_t^2}{2}\mathbb{V}(\|v(x_t)\|_2^2|x_t) \\
= & -\left(1 - \frac{S_F\gamma_t}{2}\right)\gamma_t\|\nabla F(x_t)\|_2^2 + \frac{S_F\gamma_t^2}{2}\mathbb{V}(\|v(x_t)\|_2^2|x_t) \\
\leq & -\frac{\gamma_t}{2}\|\nabla F(x_t)\|_2^2 + \frac{S_F\gamma_t^2}{2}\mathbb{V}(\|v(x_t)\|_2^2|x_t) \\
\leq & -\gamma_t\mu(F(x_t) - F(x^*)) + \frac{S_F\gamma_t^2}{2}\mathbb{V}(\|v(x_t)\|_2^2|x_t),
\end{aligned}
\tag{14}
$$

where the second to last inequality uses the assumption that $\gamma_t \leq \frac{1}{S_F}$, the last inequality uses PL condition derived from strong convexity. By strong convexity, we have

$$
\begin{aligned}
F(x^*) \geq & F(x) + \nabla F(x)^\top(x^* - x) + \frac{\mu}{2}\|x^* - x\|_2^2 \\
\geq & \min_{\bar{x}} F(x) + \nabla F(x)^\top(\bar{x} - x) + \frac{\mu}{2}\|\bar{x} - x\|_2^2 \\
= & F(x) - \frac{1}{2\mu}\|\nabla F(x)\|_2^2.
\end{aligned}
\tag{15}
$$

Therefore we have the PL condition:

$$
\frac{1}{2\mu}\|\nabla F(x)\|_2^2 \geq F(x) - F(x^*).
$$

Subtracting $F(x^*)$ on both sides of (14) and taking full expectation, we have

$$
\begin{aligned}
\mathbb{E}[F(x_{t+1}) - F(x^*)] \leq & (1 - \gamma_t\mu)\mathbb{E}[F(x_t) - F(x^*)] + \frac{S_F\gamma_t^2}{2}\mathbb{V}(\|v(x_t)\|_2^2) \\
\leq & (1 - \gamma_t\mu)\mathbb{E}[F(x_t) - F(x^*)] + \frac{S_F\gamma_t^2}{2}V
\end{aligned}
\tag{16}
$$

For fixed $\gamma_t = \gamma \leq \frac{1}{S_F}$, by induction, we have

$$
\mathbb{E}[F(x_t) - F(x^*)] \leq (1 - \gamma\mu)^{t-1}\mathbb{E}[F(x_1) - F(x^*)] + \frac{S_F\gamma V}{2\mu}.
$$

It holds for $t = 1$. Suppose that it holds true for $t$.

$$
\begin{aligned}
\mathbb{E}[F(x_{t+1}) - F(x^*)] \leq & (1 - \gamma\mu)\left((1 - \gamma\mu)^{t-1}\mathbb{E}[F(x_1) - F(x^*)] + \frac{S_F\gamma V}{2\mu}\right) + \frac{S_F\gamma^2}{2}V \\
= & (1 - \gamma\mu)^t\mathbb{E}[F(x_1) - F(x^*)] + \frac{S_F\gamma V}{2\mu}.
\end{aligned}
$$

It completes the induction. Since $x_1$ and $x^*$ are deterministic, we have

$$
\mathbb{E}[F(x_T) - F(x^*)] \leq (1 - \gamma\mu)^{T-1}[F(x_1) - F(x^*)] + \frac{S_F\gamma V}{2\mu}.
$$

**Strongly Convex case (a2)** Denote $a_t = \frac{1}{2}\mathbb{E}\|x_t - x^*\|_2^2$.

$$
\begin{aligned}
a_{t+1} = & \frac{1}{2}\mathbb{E}\|x_{t+1} - x^*\|_2^2 \\
= & \frac{1}{2}\mathbb{E}\|x_t - \gamma_t v(x_t) - x^*\|_2^2 \\
= & a_t + \frac{1}{2}\gamma_t^2\mathbb{E}\|v(x_t)\|_2^2 - \gamma_t\mathbb{E}v(x_t)^\top(x_t - x^*).
\end{aligned}
\tag{17}
$$

Following (17), we have

$$a_{t+1} = a_t + \frac{1}{2}\gamma_t^2 \mathbb{E}\|v(x_t)\|_2^2 - \gamma_t \mathbb{E}v(x_t)^\top(x_t - x^*)$$

$$\leq a_t + \frac{1}{2}\gamma_t^2 \mathbb{V}(v(x_t)) + \frac{1}{2}\gamma_t^2 \mathbb{E}\|\nabla F(x_t)\|_2^2 - \gamma_t(\mathbb{E}F(x_t) - F(x^*) + \frac{\mu}{2}\mathbb{E}\|x_t - x^*\|_2^2)$$

$$\leq a_t + \frac{1}{2}\gamma_t^2 \mathbb{V}(v(x_t)) + \frac{1}{2}\gamma_t^2 S_F^2 \mathbb{E}\|x_t - x^*\|_2^2 - \gamma_t(\mu\mathbb{E}\|x_t - x^*\|_2^2)$$

$$= (1 - 2\mu\gamma_t + \gamma_t^2 S_F^2)a_t + \frac{1}{2}\gamma_t^2 \mathbb{V}(v(x_t)),$$

where the first inequality uses the decomposition of mean squared error and strong convexity, the second inequality uses the smoothness of $F$ and that $\nabla F(x^*) = 0$.

To prove (a2), since $z = 2S_F^2/\mu^2$ and $\gamma_t = \frac{1}{\mu(t+z)}$, we have $\gamma_t^2 S_F^2 \leq 0.5\mu\gamma_t$. As a result, we have

$$\mathbb{E}\|x_{t+1} - x^*\|_2^2 \leq (1 - 1.5\mu\gamma_t)\mathbb{E}\|x_t - x^*\|_2^2 + \gamma_t^2 \mathbb{V}(v(x_t))$$

$$\leq \left(1 - \frac{1.5}{t+z}\right)\|x_t - x^*\|_2^2 + \frac{V}{\mu^2(t+z)^2}.$$

By induction, one has

$$\mathbb{E}\|x_t - x^*\|_2^2 \leq \frac{2\max\{V, \mu^2(1+z)\|x_1 - x^*\|_2^2\}}{\mu^2(t+z)}.$$

It holds for $t = 1$. Suppose that it holds for $t$, by recursion, we have

$$\mathbb{E}\|x_{t+1} - x^*\|_2^2 \leq \left(1 - \frac{1.5}{t+z}\right)\mathbb{E}\|x_t - x^*\|_2^2 + \frac{V}{\mu^2(t+z)^2}$$

$$\leq \left(1 - \frac{1.5}{t+z}\right)\frac{2\max\{V, \mu^2(1+z)\|x_1 - x^*\|_2^2\}}{\mu^2(t+z)} + \frac{V}{\mu^2(t+z)^2}$$

$$= \frac{(2(t+z)^2 - (t+z+3))\max\{V, \mu^2(1+z)\|x_1 - x^*\|_2^2\} + V(t+z+1)}{\mu^2(t+z)^2(t+z+1)}$$

$$\leq \frac{2(t+z)^2 \max\{V, \mu^2(1+z)\|x_1 - x^*\|_2^2\}}{\mu^2(t+z)^2(t+z+1)}$$

$$= \frac{2\max\{V, \mu^2(1+z)\|x_1 - x^*\|_2^2\}}{\mu^2(t+z+1)}.$$

The last inequality holds as $V \leq \max\{V, \mu^2(1+z)\|x_1 - x^*\|_2^2\}$. It completes the induction. Since $F$ is $S_F$-smooth and that $\nabla F(x^*) = 0$, one has that

$$\mathbb{E}F(x_T) - F(x^*) \leq \nabla F(x^*)^\top \mathbb{E}(x_T - x^*) + \frac{S_F}{2}\mathbb{E}\|x_T - x^*\|_2^2 \leq \frac{S_F \max\{V, \mu^2(1+z)\|x_1 - x^*\|_2^2\}}{\mu^2(T+z)}.$$

**Convex Case (b).** We prove the case when $F(x)$ is convex. Dividing $\gamma_t$ on both sides of (17), it holds that

$$\mathbb{E}v(x_t)^\top(x_t - x^*) \leq \frac{a_t - a_{t+1}}{\gamma_t} + \frac{1}{2}\gamma_t \mathbb{E}\|v(x_t)\|_2^2.$$

Since $F(x)$ is convex and $v(x_t)$ is an unbiased gradient estimator of $F(x_t)$ conditioned on $x_t$, we have

$$-\mathbb{E}v(x_t)^\top(x_t - x^*) = -\nabla F(x_t)^\top(x_t - x^*) \leq F(x^*) - F(x_t).$$

Summing up the two inequalities above and rearranging, it holds that conditioned on $x_t$,

$$F(x_t) - F(x^*) \leq \frac{a_t - a_{t+1}}{\gamma_t} + \frac{1}{2}\gamma_t \mathbb{E}\|v(x_t)\|_2^2$$

$$\leq \frac{a_t - a_{t+1}}{\gamma_t} + \frac{1}{2}\gamma_t \mathbb{V}(v(x_t)) + \frac{1}{2}\gamma_t\|\nabla F(x_t)\|_2^2$$

$$= \frac{a_t - a_{t+1}}{\gamma_t} + \frac{1}{2}\gamma_t \mathbb{V}(v(x_t)) + \frac{1}{2}\gamma_t\|\nabla F(x_t) - \nabla F(x^*)\|_2^2$$

$$\leq \frac{a_t - a_{t+1}}{\gamma_t} + \frac{1}{2}\gamma_t \mathbb{V}(v(x_t)) + \gamma_t S_F[F(x_t) - F(x^*)].$$

The last inequality holds as $F$ is convex and $S_F$-smooth on $\mathbb{R}^d$. Equivalently, one has that for any $x, y \in \mathbb{R}^d$,

$$F(x) - F(y) \geq \nabla F(y)^\top (x - y) + \frac{1}{2S_F} \|\nabla F(x) - \nabla F(y)\|_2^2.$$

It implies that

$$\mathbb{E}[F(x_t) - F(x^*)] \leq \frac{1}{1 - \gamma_t S_F} \left[ \frac{a_t - a_{t+1}}{\gamma_t} + \frac{1}{2} \gamma_t \mathbb{V}(v(x_t)) \right].$$

For $\gamma_t \leq \frac{1}{2S_F}$, it holds that $\frac{1}{1 - \gamma_t S_F} \leq 2$. By definition of $\widehat{x}_T$, Jensen's inequality, and convexity of $F$, it holds that

$$\mathbb{E}[F(\widehat{x}_T) - F(x^*)] \leq \frac{1}{T} \sum_{t=1}^T \mathbb{E}[F(x_t) - F(x^*)]$$

$$\leq \frac{1}{T} \sum_{t=1}^T \gamma_t \mathbb{V}(v(x_t)) + \frac{1}{T} \sum_{t=2}^T \|x_t - x^*\|_2^2 \left( \frac{1}{\gamma_t} - \frac{1}{\gamma_{t-1}} \right) + \frac{1}{\gamma_1 T} \|x_1 - x^*\|_2^2 \qquad (18)$$

$$\leq \gamma V + \frac{\|x_1 - x^*\|_2^2}{\gamma T}.$$

where the last inequality holds as $\gamma_t = \gamma$.

**Nonconvex smooth case (c)**   Next we prove the case when $F(x)$ is $S_F$-smooth but nonconvex on $\mathbb{R}^d$. Since $F(x)$ is $S_F$-smooth, it holds that

$$F(x_{t+1}) - F(x_t) \leq \nabla F(x_t)^\top (x_{t+1} - x_t) + \frac{S_F}{2} \|x_{t+1} - x_t\|_2^2$$

$$= -\nabla F(x_t)^\top \gamma_t v(x_t) + \frac{S_F \gamma_t^2}{2} \|v(x_t)\|_2^2.$$

Taking expectation on both side, it holds

$$\mathbb{E}[F(x_{t+1}) - F(x_t)]$$

$$\leq -\gamma_t \mathbb{E}\|\nabla F(x_t)\|_2^2 + \frac{S_F \gamma_t^2}{2} \mathbb{E}\|v(x_t)\|_2^2$$

$$= -\gamma_t \mathbb{E}\|\nabla F(x_t)\|_2^2 + \frac{S_F \gamma_t^2}{2} \mathbb{E}\|\nabla F(x_t)\|_2^2 + \frac{S_F \gamma_t^2}{2} \mathbb{V}(v(x_t)).$$

Summing up from $t = 1$ to $t = T$, taking full expectation, and setting $\gamma_t = \gamma \leq 1/S_F$, we have

$$\mathbb{E}[F(x_{T+1}) - F(x_1)]$$

$$\leq (-\gamma + \frac{S_F \gamma^2}{2}) \sum_{t=1}^T \mathbb{E}\|\nabla F(x_t)\|_2^2 + \sum_{t=0}^{T-1} \frac{S_F \gamma^2}{2} \mathbb{V}(v(x_t))$$

$$\leq -\frac{\gamma}{2} \sum_{t=1}^T \mathbb{E}\|\nabla F(x_t)\|_2^2 + \sum_{t=1}^T \frac{S_F \gamma^2}{2} \mathbb{V}(v(x_t)).$$

As a result, it holds that

$$\mathbb{E}\|\nabla F(\widehat{x}_T)\|_2^2 \leq \frac{1}{T} \sum_{t=1}^T \mathbb{E}\|\nabla F(x_t)\|_2^2 \leq \frac{2\mathbb{E}(F(x_1) - F(x_{T+1}))}{\gamma T} + S_F \gamma V$$

$$\leq \frac{2(F(x_1) - F(x^*))}{\gamma T} + S_F \gamma V.$$

$\square$

# B  Bias, Variance and Cost of Gradient Estimators

This section demonstrates the bias, variance, and per-iteration cost of the $L$-SGD and the MLMC-based gradient estimators. The simplified results are given in the main content in Table 2. We use RT-MLMC to illustrate how to select the batch size for V-MLMC and the probability distribution for randomized MLMC methods.

The following lemma formally characterizes the expectation of $L$-SGD and MLMC-based gradient estimators.

**Lemma B.1.** *For any $x \in \mathbb{R}^d$ independent of*

$$\mathbb{E}v^{\mathrm{L-SGD}}(x) = \mathbb{E}v^{\mathrm{V-MLMC}}(x) = \mathbb{E}v^{\mathrm{RT-MLMC}}(x) = \nabla F^L(x).$$

*If additionally Assumption 2.2 holds, we further have*

$$\mathbb{E}v^{\mathrm{RU-MLMC}}(x) = \mathbb{E}v^{\mathrm{RR-MLMC}}(x) = \nabla F(x).$$

*Proof.* The proof for $L$-SGD is straightforward. For V-MLMC, it holds that

$$\mathbb{E}v^{\mathrm{V-MLMC}}(x) = \mathbb{E}\left[ \sum_{l=0}^{L} \frac{1}{n_l} \sum_{i=1}^{n_l} H^l(x, \zeta_i^l) \right] = \sum_{l=0}^{L}[\nabla F^l(x) - \nabla F^{l-1}(x)] = \nabla F^L(x).$$

For RT-MLMC, we have

$$\mathbb{E}v^{\mathrm{RT-MLMC}}(x) = \mathbb{E}\left[ \frac{H^l(x, \zeta^l)}{q_l} \right] = \sum_{l=0}^{L} q_l \frac{\nabla F^l(x) - \nabla F^{l-1}(x)}{q_l} = \nabla F^L(x).$$

For RU-MLMC, letting $L \to \infty$, we have

$$\mathbb{E}v^{\mathrm{RU-MLMC}}(x) = \mathbb{E}\left[ \frac{H^l(x, \zeta^l)}{q_l} \right] = \sum_{l=0}^{\infty} q_l \frac{\nabla F^l(x) - \nabla F^{l-1}(x)}{q_l} = \lim_{L \to \infty} \nabla F^L(x) = \nabla F(x),$$

where the last equality uses Assumption 2.2.

As for the RR-MLMC estimator, we have

$$\mathbb{E}v^{\mathrm{RR-MLMC}} = \mathbb{E}\sum_{l=0}^{L} p_l H^l(x, \zeta^l) = \mathbb{E}_L \sum_{l=0}^{L} p_l[\nabla F^l(x) - \nabla F^{l-1}(x)]$$

$$= \sum_{L=0}^{\infty} q_L \left( \sum_{l=0}^{L} p_l[\nabla F^l(x) - \nabla F^{l-1}(x)] \right) = \sum_{L=0}^{\infty} \sum_{l=0}^{L} q_L p_l[\nabla F^l(x) - \nabla F^{l-1}(x)]\mathbb{I}\{l \le L\}$$

$$= \sum_{l=0}^{\infty}[\nabla F^l(x) - \nabla F^{l-1}(x)] \sum_{L=l}^{\infty} q_L p_l = \sum_{l=0}^{\infty}[\nabla F^l(x) - \nabla F^{l-1}(x)] \frac{\sum_{L=l}^{\infty} q_L}{1 - \sum_{l'=0}^{l-1} q_{l'}}$$

$$= \sum_{l=0}^{\infty}[\nabla F^l(x) - \nabla F^{l-1}(x)] = \lim_{L \to \infty} \nabla F^L(x)$$

$$= \nabla F(x).$$

By convention, we let $\sum_{l'=0}^{-1} q_{l'} = 0$ and $\nabla F^{-1}(x) = 0$. Similar to RU-MLMC, the last equality uses Assumption 2.2. $\square$

In the following, we demonstrate the variance and the cost of those estimators. Recall that we use $A$ to denote a method, where $A$ can be either of $L$-SGD, V-MLMC, RT-MLMC, RU-MLMC, and RR-MLMC; $C_{\mathrm{iter}}^A$ to denote the (expected) per-iteration cost of method $A$; $\mathbb{V}(v^A)$ to denote the variance of the gradient estimator of $A$; $T^A$ to denote the iteration complexity of $A$ for achieving $\epsilon$-optimality or $\epsilon$-stationarity; $C := T^A C_{\mathrm{iter}}^A$ to denote the (expected) total cost.

Note that the per-iteration cost $C_{\mathrm{iter}}^A$ depends on different gradient constructions. By Lemma A.1, the iteration complexity $T^A$ depends on the desired accuracy $\epsilon$ and $\mathbb{V}(v^A)$. To upper bound the (expected) total cost $C$, we need first to make sure that both the per-iteration cost $C_{\mathrm{iter}}^A$ and the variance of the gradient estimator $V(v^A)$ are finite.

**Lemma B.2.** *Under Assumption 2.1, denote* $R^L(\alpha) := \sum_{l=0}^{L} 2^{\alpha l} = \frac{1-2^{\alpha(L+1)}}{1-2^{\alpha}}$ *with* $\alpha \in \mathbb{R}$ *and* $\alpha \neq 0$, *for any* $x \in \mathbb{R}^d$, *we have the following results.*

- *For* $v^{\mathrm{L-SGD}}$ *with batch size* $n_L$, *its variance and per-iteration cost satisfy the followings:*

$$\mathbb{V}(v^{\mathrm{L-SGD}}(x)) \leq \frac{\sigma^2}{n_L}.$$

$$C_{\mathrm{iter}}^{\mathrm{L-SGD}} = n_L C_L \leq n_L M_c 2^{cL}.$$

- *For* $v^{\mathrm{V-MLMC}}(x)$, *setting* $n_l = \lceil 2^{-(b+c)l/2} N \rceil$ *for a constant* $N > 0$, *its variance and per-iteration cost satisfy the followings:*

$$\mathbb{V}(v^{\mathrm{V-MLMC}}(x)) \leq \sum_{l=0}^{L} \frac{V_l}{n_l} \leq \begin{cases} M_b R^L(\frac{c-b}{2}) N^{-1} & \text{if } c \neq b; \\ M_b (L+1) N^{-1} & \text{if } c = b. \end{cases}$$

$$C_{\mathrm{iter}}^{\mathrm{V-MLMC}} = \sum_{l=0}^{L} C_l n_l \leq \begin{cases} M_c R^L(\frac{c-b}{2}) N + M_c R^L(c) & \text{if } c \neq b; \\ M_c (L+1) N + M_c R^L(c) & \text{if } c = b. \end{cases}$$

- *For* $v^{\mathrm{RT-MLMC}}(x)$, *setting* $q_l = 2^{-(b+c)l/2} R^L(-\frac{b+c}{2})^{-1}$ *so that* $\sum_{l=0}^{L} q_l = 1$, *we have the followings:*

$$\mathbb{V}(v^{\mathrm{RT-MLMC}}(x)) \leq \sum_{l=0}^{L} \frac{V_l}{q_l} = \begin{cases} M_b R^L(\frac{c-b}{2}) R^L(-\frac{b+c}{2}) & \text{if } c \neq b; \\ M_b (L+1) R^L(-\frac{b+c}{2}) & \text{if } c = b. \end{cases}$$

$$C_{\mathrm{iter}}^{\mathrm{RT-MLMC}} = \sum_{l=0}^{L} C_l q_l \leq \begin{cases} M_c R^L(\frac{c-b}{2}) R^L(-\frac{b+c}{2})^{-1} & \text{if } c \neq b; \\ M_c (L+1) R^L(-\frac{b+c}{2})^{-1} & \text{if } c = b. \end{cases}$$

- *For* $v^{\mathrm{RU-MLMC}}(x)$, *when* $c \geq b$, *either its expected per-iteration cost or its variance is unbounded. When* $c < b$, *setting* $q_l = 2^{-(b+c)l/2} R^{\infty}(-\frac{b+c}{2})^{-1}$ *so that* $\sum_{l=0}^{\infty} q_l = 1$, *we have the followings:*

$$\mathbb{V}(v^{\mathrm{RU-MLMC}}(x)) \leq \sum_{l=0}^{\infty} \frac{V_l}{q_l} = M_b R^{\infty}(\frac{c-b}{2}) R^{\infty}(-\frac{b+c}{2}).$$

$$C_{\mathrm{iter}}^{\mathrm{RU-MLMC}} = \sum_{l=0}^{\infty} C_l q_l \leq M_c R^{\infty}(\frac{c-b}{2}) R^{\infty}(-\frac{c+b}{2})^{-1}.$$

- *For* $v^{\mathrm{RR-MLMC}}(x)$, *when* $c \geq b$, *either its expected per-iteration cost or its variance is unbounded. When* $c < b$, *setting* $q_L = 2^{-(b+c)l/2}(1 - 2^{-(b+c)/2})$ *so that* $\sum_{L=0}^{\infty} q_L = 1$, *we have the followings:*

$$\mathbb{V}(v^{\mathrm{RR-MLMC}}(x)) \leq \sum_{L=0}^{\infty} q_L \Big( \sum_{l=0}^{L} p_l^2 V_l \Big) = M_b R^{\infty}(\frac{c-b}{2}).$$

$$C_{\mathrm{iter}}^{\mathrm{RR-MLMC}} = \sum_{L=0}^{\infty} q_L \Big( \sum_{l=0}^{L} C_l \Big) \leq \frac{M_c 2^c}{2^c - 1} R^{\infty}(\frac{c-b}{2}) R^{\infty}(-\frac{c+b}{2})^{-1}.$$

*Remark* B.1. Note that $R^L(\alpha) = \mathcal{O}(1)$ when $\alpha < 0$, $R^L(\alpha) = \mathcal{O}(L)$ when $\alpha = 0$, and $R^L(\alpha) = \mathcal{O}(2^{\alpha(L+1)})$ when $\alpha > 0$, $R^{\infty}(\alpha) = \mathcal{O}(1)$ when $\alpha < 0$. Lemma B.2 suggests that when $c < b$, the variance and the per-iteration cost of RT-MLMC, RU-MLMC, and RR-MLMC are all $\mathcal{O}(1)$.

In the following, we prove Lemma B.2.

*Proof.* We use the following equality to prove the results.

$$R^L(\alpha =) \sum_{l=0}^{L} 2^{\alpha l} = \frac{1 - 2^{\alpha(L+1)}}{1 - 2^{\alpha}} \text{ for } \alpha \neq 0; \quad R^{\infty}(\alpha) = (1 - 2^{\alpha}) \text{ for } \alpha < 0. \tag{19}$$

**$L$-SGD**  The statement follows directly from Assumption 2.1 and the fact that $\{\nabla h^L(x, \zeta_i^L)\}_{i=1}^{n_L}$ are independent for any given $L$.

**Vanilla MLMC**  Notice that $\{H^l(x, \zeta_i^l)\}_{i=1}^{n_l}$ are independent for any given $l$ and $H^l(x, \zeta_i^l)$ with different $l$ are independently generated. Using (19) and that fact that $n_l = \lceil 2^{-(b+c)l/2}N \rceil \in [2^{-(b+c)l/2}N, 2^{-(b+c)l/2}N + 1)$, we have the following results.

$$\mathbb{V}(v^{\text{V}-\text{MLMC}}(x)) \leq \sum_{l=0}^{L} \frac{V_l}{n_l} \leq M_b \left[ \sum_{l=0}^{L} \frac{2^{-bl}}{2^{-(b+c)l/2}} \right] N^{-1} = \begin{cases} M_b \frac{1-2^{-(b-c)(L+1)/2}}{1-2^{-(b-c)/2}} N^{-1} & \text{if } c \neq b; \\[2mm] M_b(L+1)N^{-1} & \text{if } c = b. \end{cases}$$

$$C_{\text{iter}}^{\text{V}-\text{MLMC}} = \sum_{l=0}^{L} n_l C_l \leq M_c \left[ \sum_{l=0}^{L} 2^{cl} 2^{-(b+c)l/2} \right] N + M_c \sum_{l=0}^{L} 2^{cl}$$
$$= \begin{cases} M_c \frac{1-2^{-(b-c)(L+1)/2}}{1-2^{-(b-c)/2}} N + M_c \frac{2^{c(L+1)}-1}{2^c-1} & \text{if } c \neq b; \\[2mm] M_c(L+1)N + M_c \frac{2^{c(L+1)}-1}{2^c-1} & \text{if } c = b. \end{cases}$$

**Randomized Truncated MLMC**  Using (19) and the statement is obvious by simple calculation.

**Randomized Unbiased MLMC**  To make sure that $C_{\text{iter}}^{\text{RU}-\text{MLMC}} = \sum_{l=0}^{\infty} C_l q_l < \infty$, it requires that the following inequality holds for large $l$

$$q_l 2^{cl} < 1 \implies q_l < 2^{-cl}.$$

To make sure that $\mathbb{V}(v^{\text{RU}-\text{MLMC}}(x)) = \sum_{l=0}^{\infty} V_l/q_l < \infty$, it requires for large enough $l$ such that

$$\frac{2^{-bl}}{q_l} < 1 \implies q_l > 2^{-bl}.$$

As a result, to make sure that both the expected per-iteration cost and the variance are finite, the following inequality should hold for large $l$

$$2^{-bl} < q_l < 2^{-cl}.$$

The inequality holds only when $c < b$. Otherwise, either the variance or the expected per-iteration cost is unbounded for any selection of $\{q_l\}_{l=0}^{\infty}$.

When $c < b$, plugging in $q_l$, $V_l$ and $C_l$, using (19) and letting $L \to \infty$, we have the upper bounds on the variance and the expected per-iteration cost of the randomized unbiased MLMC gradient estimator.

**Russian Roulette MLMC**  The expected cost to generate such an estimator is

$$C_{\text{iter}}^{\text{RR}-\text{MLMC}} = \mathbb{E}_L \sum_{l=0}^{L} C_l = \sum_{L=0}^{\infty} q_L \sum_{l=0}^{L} C_l$$
$$\leq \sum_{L=0}^{\infty} q_L \sum_{l=0}^{L} M_c 2^{cl} = M_c \sum_{L=0}^{\infty} q_L \frac{2^{c(L+1)}-1}{2^c-1}.$$

To make sure that the expected per-iteration cost is finite, it requires that $q_L < 2^{-cL}$. Without loss of generality, we assume that $q_l = 2^{\alpha l}(1-2^\alpha)$ for a constant $0 < \alpha < -c$ and $l \in \mathbb{N}$ so that $\sum_{l=0}^{\infty} q_l = 1$. As a result, we have $\sum_{l'=0}^{l-1} q_{l'} = 1 - 2^{\alpha l}$. The variance of the estimator is

$$\mathbb{V}(v^{\text{RR}-\text{MLMC}}(x)) \leq \sum_{L=0}^{\infty} q_L \left( \sum_{l=0}^{L} p_l^2 V_l \right) = \sum_{L=0}^{\infty} q_L \left( \sum_{l=0}^{L} \frac{1}{(1-\sum_{l'=0}^{l-1} q_{l'})^2} \mathbb{I}\{l \leq L\} V_l \right)$$
$$= \sum_{L=0}^{\infty} \sum_{l=0}^{L} q_L \frac{1}{(1-\sum_{l'=0}^{l-1} q_{l'})^2} \mathbb{I}\{l \leq L\} V_l$$

$$= \sum_{l=0}^{\infty} V_l \frac{\sum_{L=l}^{\infty} q_L}{(1 - \sum_{l'=0}^{l-1} q_{l'})^2}$$

$$= \sum_{l=0}^{\infty} \frac{V_l}{1 - \sum_{l'=0}^{l-1} q_{l'}}.$$

Note that we abuse of notation $\sum_{l'=0}^{l-1} q_{l'} = 0$ when $l = 0$. To make sure that the variance is bounded, it requires that

$$\frac{2^{-bl}}{1 - \sum_{l'=0}^{l-1} q_{l'}} < 1 \implies 2^{-bl} < 1 - (1 - 2^{\alpha l}),$$

It further implies that

$$2^{-bl} < 2^{\alpha l} \implies \alpha > -b.$$

Therefore, to ensure that both the expected per-iteration cost and the variance are finite, it requires that $c < b$. Selecting $\alpha = -(b+c)/2$, we have

$$\mathbb{V}(v^{\mathrm{RR-MLMC}}(x)) \leq \sum_{l=0}^{\infty} M_b 2^{-bl}/2^{-(b+c)l/2} = M_b \frac{1}{1 - 2^{-(b-c)/2}}.$$

$$C_{\mathrm{iter}}^{\mathrm{RR-MLMC}} \leq M_c \sum_{L=0}^{\infty} q_L \frac{2^{c(L+1)}}{2^c - 1} = M_c \frac{2^c}{2^c - 1} (1 - 2^{-(b+c)/2}) \sum_{L=0}^{\infty} 2^{(c-b)L/2}$$

$$= M_c \frac{2^c}{2^c - 1} \frac{1 - 2^{-(b+c)/2}}{1 - 2^{-(b-c)/2}}.$$

$\square$

# C   Total Cost Analysis

This section discusses the (expected) total cost of $L$-SGD, V-MLMC, RT-MLMC, RU-MLMC, and RR-MLMC. We use $V(v^A)$ to denote the uniform upper bound on the variance of the gradient estimator constructed by Algorithm $A$. Let $\hat{x}_T$ be uniformly selected from $x_1, ..., x_T$.

In the (strongly) convex case, by error decomposition (9) and Assumption 2.1(a), the expected error of $L$-SGD, V-MLMC, RT-MLMC satisfy that

$$\mathbb{E}F(\hat{x}^A) - F(x^*) \leq 2B_L + \mathbb{E}F^L(\hat{x}^A) - F^L(x^L). \tag{20}$$

To make sure that the expected error is bounded by $\epsilon$, we set $L = \lceil 1/a \log(4M_a/\epsilon) \rceil$ so that $2B_L = 2M_a 2^{-aL} \leq \epsilon/2$. What remains is to use the convergence results of SGD, i.e., Lemma A.1 in Appendix A, to show that $\mathbb{E}F^L(\hat{x}^A) - F^L(x^L) \leq \epsilon/2$. Note that under the choice of $L$, we have $2^{cL} \leq 2^c (4M_a/\epsilon)^{c/a}$.

In the nonconvex smooth case, by Assumption 2.2, we have

$$\mathbb{E}\|\nabla F(\hat{x}_T^A)\|_2^2 \leq 2\mathbb{E}\|\nabla F^L(\hat{x}_T^A)\|_2^2 + 2\mathbb{E}\|\nabla F(\hat{x}_T^A) - \nabla F^L(\hat{x}_T^A)\|_2^2$$
$$\leq 2\mathbb{E}\|\nabla F^L(\hat{x}_T^A)\|_2^2 + 2M_a 2^{-aL}. \tag{21}$$

To make sure that $\mathbb{E}\|\nabla F(\hat{x}_T)\|_2^2 \leq \epsilon^2$, we set $L = \lceil 1/a \log(4M_a/\epsilon^2) \rceil$ so that $2M_a 2^{-aL} \leq \epsilon^2/2$. What remains is to use Lemma A.1(d) to show that $2\mathbb{E}\|\nabla F^L(\hat{x}_T)\|_2^2 \leq \epsilon^2/2$.

By Lemma B.2, the variance of the gradient estimator of $L$-SGD with batch size $n_L = \mathcal{O}(1)$ is $\sigma^2/n_L = \mathcal{O}(1)$. The variance of the gradient estimator of RT-MLMC is $\mathcal{O}(1)$ when $c < b$, $\mathcal{O}(\log(\epsilon^{-1}))$ when $c = b$, and $\mathcal{O}(\epsilon^{-(b+c)/2a})$ when $c > b$. On the contrary, as we have mentioned in Remark 4, V-MLMC has to use large mini-batches, thus the variance is very small and of order $\mathcal{O}(\epsilon^{-1})$.

When the variance of the gradient estimator is at least $\mathcal{O}(1)$, we demonstrate the relationship among the variance of the gradient estimator, the (expected) cost to construct the gradient estimator, and the (expected) total cost for achieving $\epsilon$-optimality in the convex setting ($\epsilon$-stationarity in the nonconvex smooth setting respectively) via the following theorem.

**Theorem C.1.** *For $\epsilon > 0$ small enough, let $\rho_0 > 0$ be a constant that does not depend on $\epsilon$. If $V(v^A) \geq \rho_0$, we have the followings.*

- *If Assumption 2.1 holds and $F^L$ is $\mu$-strongly convex for $L = \lceil 1/a \log(4M_a/\epsilon) \rceil$, set stepsizes $\gamma_t = \frac{1}{\mu(t+2S_F^2/\mu^2)}$. Let $x_T^A$ be the $T$-th iteration of $A$, $x_T^A$ is an $\epsilon$-optimal solution of $F$ if*

$$T \geq \frac{2S_F \max\{V(v^A), \mu^2(1 + 2S_F^2/\mu^2)\|x_1 - x^L\|_2^2\}}{\mu^2 \epsilon}.$$

*The (expected) total cost of Algorithm $A$ for finding $x_T^A$ can be upper bounded by*

$$C \leq 4C_{\text{iter}}^A S_F \max\{V(v^A), \mu^2(1 + 2S_F^2/\mu^2)\|x_1 - x^L\|_2^2\}\mu^{-2}\epsilon^{-1}.$$

- *If Assumption 2.1 holds and $F^L$ is convex for $L = \lceil 1/a \log(4M_a/\epsilon) \rceil$, set stepsizes $\gamma_t = \frac{1}{\sqrt{TV(v^A)}}$. If it holds that*

$$T \geq 4V(v^A)(1 + \|x_1 - x^L\|_2^2)^2 \epsilon^{-2},$$

*then $\gamma_t \leq 1/2S_F$ and $\widehat{x}_T^A$, selected uniformly from the first $T$ iteration of $A$, is an $\epsilon$-optimal solution of $F$. The (expected) total cost of Algorithm $A$ for finding $\widehat{x}_T^A$ is upper bounded by*

$$C \leq 8C_{\text{iter}}^A V(v^A)(1 + \|x_1 - x^L\|_2^2)^2 \epsilon^{-2}.$$

- *If Assumptions 2.1(b)(c)(d) and 2.2 holds set $L = \lceil 1/a \log(4M_a/\epsilon^2) \rceil$ and stepsizes $\gamma_t = \frac{1}{\sqrt{TV(v^A)}}$. If $T$ satisfies*

$$T \geq 16V(v^A)(2(F^L(x_1) - F^L(x^L)) + S_F)^2 \epsilon^{-4},$$

*then $\gamma_t \leq 1/(S_F)$ and $\widehat{x}_T^A$ is an $\epsilon$-stationarity point of $F$. The (expected) total cost of Algorithm $A$ for finding $\widehat{x}_T^A$ is upper bounded by*

$$C \leq 32C_{\text{iter}}^A V(v^A)(2(F^L(x_1) - F^L(x^L)) + S_F)^2 \epsilon^{-4}.$$

*Remark* C.1. For $V(v^A) \geq \rho_0$, the condition on the stepsizes $\gamma_t = 1/\sqrt{TV(v^A)} \leq 1/(2S_F)$ in Lemma A.1(b) and $\gamma_t = 1/\sqrt{TV(v^A)} \leq 1/S_F$ in Lemma A.1(c) can be easily satisfied as long as $\epsilon$ is small enough.

*Proof.* In (strongly) convex case, by (20), setting $L = \lceil 1/a \log(4M_a/\epsilon) \rceil$, we have $M_c 2^{cL} \leq M_c 2^c (4M_a/\epsilon)^{c/a}$ and it remains to show $\mathbb{E}F^L(x_T^A) - F^L(x^L) \leq \epsilon/2$.

**Strongly convex case** By Lemma A.1(a2), replacing $x^*$ with $x^L$, $F$ with $F^L$, and $V$ with $V(v^A)$, we have

$$\mathbb{E}[F^L(x_T^A) - F^L(x^L)] \leq \frac{S_F \max\{V(v^A), \mu^2(1 + 2S_F^2/\mu^2)\|x_1 - x^L\|_2^2\}}{\mu^2(T + 2S_F^2/\mu^2)}.$$

To guarantee that $x_T$ is $\epsilon$-optimal solution of $F$, we let the right-hand-side of the previous inequality to be smaller or equal to. Therefore, we need

$$T \geq \frac{2S_F \max\{V(v^A), \mu^2(1 + 2S_F^2/\mu^2)\|x_1 - x^L\|_2^2\}}{\mu^2 \epsilon}.$$

Selecting the smallest $T^* \in \mathbb{N}$ such that the requirement on $T$ is satisfied, since $\epsilon$ is small, the total cost $C$ satisfies:

$$C = T^* C_{\text{iter}}^A \leq \frac{4C_{\text{iter}}^A S_F \max\{V(v^A), \mu^2(1 + 2S_F^2/\mu^2)\|x_1 - x^L\|_2^2\}}{\mu^2 \epsilon}.$$

**Convex case**  If $T \geq 4V(v^A)(1 + \|x_1 - x^L\|_2^2)^2\epsilon^{-2}$, we have $\gamma_t = 1/\sqrt{TV(v^A)} \leq 1/(2S_F)$. By Lemma A.1(b), plugging in stepsizes $\gamma_t = 1/\sqrt{TV(v^A)}$ and replacing $x^*$ with $x^L$, $F$ with $F^L$, and $V$ with $V(v^A)$, we have

$$\mathbb{E}[F(\widehat{x}_T^A) - F(x^L)] \leq \frac{\sqrt{V(v^A)}(1 + \|x_1 - x^L\|_2^2)}{\sqrt{T}}.$$

To make sure the right-hand side is less or equal to $\epsilon/2$, it suffices to have

$$T \geq 4V(v^A)(1 + \|x_1 - x^L\|_2^2)^2\epsilon^{-2}.$$

Selecting the smallest $T^* \in \mathbb{N}$ such that the requirement on $T$ is satisfied, the total cost $C$ satisfies:

$$C = T^* C_{\text{iter}}^A \leq 8C_{\text{iter}}^A V(v^A)(1 + \|x_1 - x^L\|_2^2)^2\epsilon^{-2}.$$

**Nonconvex smooth case**  By (21), setting $L = \lceil 1/a \log(4M_a/\epsilon^2) \rceil$, we have that $2M_a 2^{-aL} \leq \epsilon^2/2$ and $2^{cL} \leq 2^c(4M_a/\epsilon^2)^{c/a}$. If $T \geq 16V(v^A)(2(F^L(x_1) - F(x^L)) + S_F)^2\epsilon^{-4}$, we have $\gamma_t = 1/\sqrt{TV(v^A)} \leq 1/S_F$. By Lemma A.1(c), using stepsizes $\gamma_t = 1/\sqrt{TV(v^A)}$, replacing $x^*$ with $x^L$, $F$ with $F^L$, and $V$ with $V(v^A)$, we have

$$\mathbb{E}\|\nabla F^L(\widehat{x}_T^A)\|_2^2 \leq \frac{\sqrt{V(v^A)}(2(F^L(x_1) - F^L(x^L)) + S_F)}{\sqrt{T}}.$$

To make sure that the right-hand-side is bounded by $\epsilon^2/4$, it suffices to have

$$T \geq 16V(v^A)(2(F^L(x_1) - F^L(x^L)) + S_F)^2\epsilon^{-4}.$$

Using the smallest $T^*$ that satisfies the requirement on $T$, we have

$$C = T^* C_{\text{iter}}^A \leq 32C_{\text{iter}}^A V(v^A)(2(F^L(x_1) - F^L(x^L)) + S_F)^2\epsilon^{-4}.$$

$\square$

## C.1  Total Cost of L-SGD

In this subsection, we show the total cost of $L$-SGD when the batch size $n_L = 1$ using Theorem C.1.

**Theorem C.2.** *For L-SGD with batch size $n_L = 1$, if $\epsilon > 0$ is small enough, we have the followings.*

- *If Assumptions 2.1 holds and $F^L$ is $\mu$-strongly convex with $L = \lceil 1/a \log(4M_a/\epsilon) \rceil$, set stepsizes $\gamma_t = \frac{1}{\mu(t + 2S_F^2/\mu^2)}$. To ensure that $x_T^{\text{L−SGD}}$ is an $\epsilon$-optimal solution of $F$, the total cost of L-SGD satisfies*

$$C = \mathcal{O}(M_c M_a^{c/a}\epsilon^{-1-c/a}).$$

- *If Assumptions 2.1 holds and $F^L$ is convex with $L = \lceil 1/a \log(4M_a/\epsilon) \rceil$, set stepsizes $\gamma_t = 1/\sqrt{T\sigma^2}$. To ensure that $\widehat{x}_T^{\text{L−SGD}}$ is an $\epsilon$-optimal solution of $F$, the total cost of L-SGD satisfies*

$$C = \mathcal{O}(M_c M_a^{c/a}\epsilon^{-2-c/a}).$$

- *If Assumptions 2.1(b)(c)(d) and 2.2 hold, set $L = \lceil 1/a \log(4M_a/\epsilon^2) \rceil$ and stepsizes $\gamma_t = 1/\sqrt{T\sigma^2}$. To ensure that $\widehat{x}_T^{\text{L−SGD}}$ is an $\epsilon$-stationarity point of $F$, the total cost of L-SGD satisfies*

$$C = \mathcal{O}(M_c M_a^{c/a}\epsilon^{-4-2c/a}).$$

*Proof.*  By Lemma B.2, we have

$$\mathbb{V}(v^{\text{L−SGD}}) \leq V(v^{\text{L−SGD}}) = \sigma^2.$$

$$C_{\text{iter}}^{\text{L−SGD}} = n_L C_L \leq M_c 2^{cL} \leq \begin{cases} M_c 2^c(4M_a/\epsilon)^{c/a} & \text{if } L = \lceil 1/a \log(4M_a/\epsilon) \rceil; \\ M_c 2^c(4M_a/\epsilon^2)^{c/a} & \text{if } L = \lceil 1/a \log(4M_a/\epsilon^2) \rceil. \end{cases}$$

The proof is done by plugging $V(v^{\text{L−SGD}})$ and $C_{\text{iter}}^{\text{L−SGD}}$ with $n_L = 1$ and $L = \lceil 1/a \log(4M_a/\epsilon) \rceil$ ($L = \lceil 1/a \log(4M_a/\epsilon^2) \rceil$) in convex (nonconvex, respectively) setting from Lemma B.2 into Theorem C.1.

$\square$

*Remark* C.2. We mentioned in Remark 4 that L-SGD cannot use large mini-batch sizes, i.e., $n_L = \mathcal{O}(\epsilon^{-1})$, to reduce the total cost as V-MLMC does. We use the convex case for demonstration. If $n_L = \mathcal{O}(\epsilon^{-1})$, then $V(v^{L-SGD}) = \mathcal{O}(\epsilon)$. Using Lemma A.1(c) with stepsizes $\gamma_t = 1/(2S_F)$, we know that

$$\mathbb{E}F^L(\widehat{x}_T^{L-SGD}) - F^L(x^L) \leq \mathcal{O}(\epsilon) + \frac{2S_F \|x_1 - x^L\|_2^2}{T}.$$

Therefore the iteration complexity $T = \mathcal{O}(\epsilon^{-1})$. The total cost would be $Tn_L C_L = \mathcal{O}(\epsilon^{-2-c/a})$, which remains the same comparing to without using batch size, i.e., $n_L = 1$. The reason is that using large batch sizes for L-SGD only reduces the iteration complexity while using large batch sizes for V-MLMC reduces the iteration complexity and the per-iteration cost simultaneously.

## C.2  Expected Total Cost of RT-MLMC

**Theorem C.3** (Full version of Theorem 3.1). *For RT-MLMC with* $q_l = 2^{-(b+c)l/2}\big(\frac{1-2^{-(b+c)(L+1)/2}}{1-2^{-(b+c)/2}}\big)^{-1}$, *if* $\epsilon > 0$ *is small enough, we have the followings.*

- *If Assumptions 2.1 holds and $F^L$ is $\mu$-strongly convex with $L = \lceil 1/a \log(4M_a/\epsilon)\rceil$, set stepsizes $\gamma_t = \frac{1}{\mu(t+2S_F^2/\mu^2)}$. To ensure that $x_T^{RT-MLMC}$ is an $\epsilon$-optimal solution of $F$, the expected total cost of RT-MLMC satisfies*

$$C = \begin{cases} \mathcal{O}(M_b M_c \epsilon^{-1}) & \text{if } c < b; \\ \mathcal{O}(M_b M_c a^{-2} \epsilon^{-1} \log^2(4M_a/\epsilon)) & \text{if } c = b; \\ \mathcal{O}(M_b M_c M_a^{(c-b)/a} \epsilon^{-1-(c-b)/a}) & \text{if } c > b. \end{cases}$$

- *If Assumptions 2.1 holds and $F^L$ is convex with $L = \lceil 1/a \log(4M_a/\epsilon)\rceil$, set stepsizes $\gamma_t = 1/\sqrt{TV(v^{RT-MLMC})}$. To ensure that $\widehat{x}_T^{RT-MLMC}$ is an $\epsilon$-optimal solution of $F$, the expected total cost of RT-MLMC satisfies*

$$C = \begin{cases} \mathcal{O}(M_b M_c \epsilon^{-2}) & \text{if } c < b; \\ \mathcal{O}(M_b M_c \epsilon^{-2} a^{-2} \log^2(4M_a/\epsilon)) & \text{if } c = b; \\ \mathcal{O}(M_b M_c M_a^{(c-b)/a} \epsilon^{-2-(c-b)/a}) & \text{if } c > b. \end{cases}$$

- *If Assumptions 2.1(b)(c)(d) and 2.2 hold, set $L = \lceil 1/a \log(4M_a/\epsilon^2)\rceil$ and stepsizes $\gamma_t = 1/\sqrt{TV(v^{RT-MLMC})}$. To ensure that $\widehat{x}_T^{RT-MLMC}$ is an $\epsilon$-stationarity point of $F$, the expected total cost of RT-MLMC satisfies*

$$C \leq \begin{cases} \mathcal{O}(M_b M_c \epsilon^{-4}) & \text{if } c < b; \\ \mathcal{O}(M_b M_c a^{-2} \log^2(4M_a/\epsilon)\epsilon^{-4}) & \text{if } c = b; \\ \mathcal{O}(M_b M_c M_a^{(c-b)/a} \epsilon^{-4-2(c-b)/a}) & \text{if } c > b. \end{cases}$$

*Proof.* Notice that for $L$ large enough and $\alpha < 0$, it holds

$$\frac{1-2^{\alpha(L+1)}}{1-2^{\alpha}} \leq \frac{1}{1-2^{\alpha}}, \frac{1-2^{\alpha}}{1-2^{\alpha(L+1)}} \leq 2(1-2^{\alpha}).$$

By Lemma B.2 for RT-MLMC, we know that $V(v^{RT-MLMC})$ is at least $\mathcal{O}(1)$ (when $c < b$). Therefore the requirement of Theorem C.1 holds. Particularly, we have

$$C_{iter}^{RT-MLMC} V(v^{RT-MLMC}) \leq \begin{cases} M_b M_c(1 - 2^{-(b+c)/2})^{-2} & \text{if } c < b; \\ 4M_b M_c a^{-2} \log^2(4M_a/\epsilon) & \text{if } c = b; \\ M_b M_c(4M_a)^{(c-b)/a} 2^{-(b-c)}(1 - 2^{-(b-c)/2})^{-2}\epsilon^{-2(c-b)/a} & \text{if } c > b. \end{cases}$$
(22)

**Strongly convex case**  In strongly convex case, when $c < b$,

$$C_{\text{iter}}^{\text{RT}-\text{MLMC}} \leq M_c \frac{1}{(1 - 2^{-(b-c)/2})(1 - 2^{-(b+c)/2})} := M_c \rho_1.$$

$$V(v^{\text{RT}-\text{MLMC}}) \leq M_b \frac{2(1 - 2^{-(b+c)/2})}{1 - 2^{-(b-c)/2}} := M_b \rho_2;$$

Plugging those into Theorem C.1, one has the desired results.

Without loss of generality, assuming that $\epsilon$ is small enough. When $c = b$, $V(v^{\text{RT}-\text{MLMC}}) = \mathcal{O}(L + 1) = \mathcal{O}(\log(1/\epsilon)) \geq \mu^2(1 + 2S_F^2/\mu^2)\|x_1 - x^L\|_2^2$, and when $c > b$ $V(v^{\text{RT}-\text{MLMC}}) = \mathcal{O}(\epsilon^{-(c-b)/2a}) \geq \mu^2(1 + 2S_F^2/\mu^2)\|x_1 - x^L\|_2^2$. By Theorem C.1, we have

$$\mathbb{E}C \leq 4C_{\text{iter}}^{\text{RT}-\text{MLMC}} S_F V(v^{\text{RT}-\text{MLMC}})\mu^{-2}\epsilon^{-1}.$$

Plugging in (22), we have the desired results.

**Convex case and nonconvex smooth case**  One needs to further verify that $\gamma_t = \frac{1}{\sqrt{TV(v^{\text{RT}-\text{MLMC}})}}$ is less or equal to $\frac{1}{2S_F}$ in the convex case and $\frac{1}{S_F}$ in the nonconvex smooth setting. Notice that $v^{\text{RT}-\text{MLMC}}$ is $\mathcal{O}(1)$ for $c < b$, $\mathcal{O}(\log(1/\epsilon))$ for $c = b$, and $\mathcal{O}(\epsilon^{-(b-c)/2a})$ for $c > b$. Therefore, the condition on the stepsize holds for large $T$ as suggested by Theorem C.1. Plugging (22) into the convex and nonconvex setting of Theorem C.1, we have the desired results. $\qquad\square$

## C.3  Total Cost of V-MLMC

In this subsection, we consider the total cost of V-MLMC.

**Theorem C.4** (Full version of Theorem 3.2). *For V-MLMC with $n_l = \lceil 2^{-(b+c)l/2}N \rceil$ for some $N > 0$, we have the followings:*

- *If Assumption 2.1 holds and $F^L$ is strongly convex, let stepsizes $\gamma_t = 1/S_F$, $L = \lceil 1/a \log(4M_a\epsilon^{-1}) \rceil$ and set $N$ as the followings:*

$$N = \begin{cases} \mathcal{O}(M_b\epsilon^{-1}\mu^{-1}) & \text{if } c < b; \\ \mathcal{O}(M_b(L + 1)\epsilon^{-1}\mu^{-1}) & \text{if } c = b; \\ \mathcal{O}(M_b 2^{-(b-c)(L+1)/2}\mu^{-1}\epsilon^{-1}) & \text{if } c > b. \end{cases}$$

*Then $x_T^{\text{V}-\text{MLMC}}$ is an $\epsilon$-optimal solution of $F$ after $T = \lceil 2\log(4[F^L(x_1) - F^L(x^L)]\epsilon^{-1})/\log(S_F/(S_F - \mu)) \rceil$ iterations. The total cost of V-MLMC satisfies*

$$C = \begin{cases} \mathcal{O}(\log(\epsilon^{-1})\epsilon^{-\{1,c/a\}}) & \text{if } c < b; \\ \mathcal{O}(\log^3(\epsilon^{-1})\epsilon^{-1} + \log(\epsilon^{-1})\epsilon^{-c/a}) & \text{if } c = b; \\ \mathcal{O}(\log(\epsilon^{-1})\epsilon^{-\max\{1+(c-b)/a,c/a\}}) & \text{if } c > b. \end{cases}$$

- *If Assumption 2.1 holds and $F^L$ is convex, set stepsizes $\gamma_t = \frac{1}{2S_F}$, $L = \lceil 1/a \log(4M_a/\epsilon) \rceil$ and set $N$ as the followings:*

$$N = \begin{cases} \mathcal{O}(M_b\epsilon^{-1}S_F^{-1}) & \text{if } c < b; \\ \mathcal{O}(M_b(L + 1)\epsilon^{-1}S_F^{-1}) & \text{if } c = b; \\ \mathcal{O}(M_b 2^{-(b-c)(L+1)/2}\epsilon^{-1}S_F^{-1}) & \text{if } c > b. \end{cases}$$

Then $\widehat{x}_T^{\mathrm{V-MLMC}}$ is an $\epsilon$-optimal solution of $F$ after $T = \lceil 8 S_F \|x_1 - x^L\|_2^2 \epsilon^{-1} \rceil$ iterations and the total cost satisfies

$$
C = \begin{cases} \mathcal{O}(\epsilon^{-1-\max\{1, c/a\}}) & \text{if } c < b; \\ \mathcal{O}(\epsilon^{-2}\log^2(\epsilon^{-1}) + \epsilon^{-1-c/a}) & \text{if } c = b; \\ \mathcal{O}(\epsilon^{-1-\max\{1+(c-b)/a, c/a\}}) & \text{if } c > b. \end{cases}
$$

- If Assumptions 2.1(b)(c)(d) and 2.2 hold, set stepsizes $\gamma_t = \frac{1}{S_F}$, $L = \lceil 1/a \log(4 M_a/\epsilon^2) \rceil$ and set $N$ as the followings:

$$
N = \begin{cases} \mathcal{O}(M_b \epsilon^{-2}) & \text{if } c < b; \\ \mathcal{O}(M_b (L+1) \epsilon^{-2}) & \text{if } c = b; \\ \mathcal{O}(M_b 2^{-(b-c)(L+1)/2} \epsilon^{-2}) & \text{if } c > b. \end{cases}
$$

Then $\widehat{x}_T^{\mathrm{V-MLMC}}$ is an $\epsilon$-stationarity point of $F$ after $\lceil 16 S_F (F^L(x_1) - F^L(x^L)) \epsilon^{-2} \rceil$ iterations and the total cost satisfies

$$
C \leq \begin{cases} \mathcal{O}(\epsilon^{-2-2\max\{1, c/a\}}) & \text{if } c < b; \\ \mathcal{O}(\log^2(\epsilon^{-1}) \epsilon^{-4} + \epsilon^{-2-2c/a}) & \text{if } c = b; \\ \mathcal{O}(\epsilon^{-2-2\max\{1+(c-b)/a, c/a\}}) & \text{if } c > b. \end{cases}
$$

*Remark* C.3. When $a \geq \min\{b, c\}$, we have $\max\{1, c/a\} = \max\{1 + (c-b)/a, c/a\} = 1$. As a result, the total cost of V-MLMC is the same as RT-MLMC when this additional condition holds.

*Proof.* In the (strongly) convex case, by (20) setting $L = \lceil 1/a \log(4 M_a/\epsilon) \rceil$, to guarantee that $x_T$ or $\widehat{x}_T$ is an $\epsilon$-optimal solution, we only need to guarantee that

$$
\mathbb{E} F^L(x_T) - F^L(x^L) \leq \epsilon/2.
$$

**Strongly convex case** By Lemma A.1(a1), we have

$$
\mathbb{E} F^L(x_T) - F^L(x^L) \leq (1 - \gamma\mu)^{T-1}[F^L(x_1) - F^L(x^L)] + \frac{S_F \gamma V(v^{\mathrm{V-MLMC}})}{2\mu}.
$$

To make sure that $(1 - \gamma\mu)^{T-1}[F^L(x_1) - F^L(x^L)] \leq \epsilon/4$ and $\frac{S_F \gamma V(v^{\mathrm{V-MLMC}})}{2\mu} \leq \epsilon/4$, it suffices to have for small $\epsilon$,

$$
T \geq 2 \log(4[F^L(x_1) - F^L(x^L)]\epsilon^{-1}) / \log(S_F/(S_F - \mu));
$$

$$
V(v^{\mathrm{V-MLMC}}) \leq \frac{\epsilon\mu}{2}.
$$

By Lemma B.2, under the condition $2^{-(b+c)l/2} N \geq 1$, we have

$$
V(v^{\mathrm{V-MLMC}}) \leq \begin{cases} M_b \frac{1 - 2^{-(b-c)(L+1)/2}}{1 - 2^{-(b-c)/2}} N^{-1} & \text{if } c \neq b; \\ M_b (L+1) N^{-1} & \text{if } c = b, \end{cases}
$$

$$
C_{\mathrm{iter}}^{\mathrm{V-MLMC}} \leq \begin{cases} M_c \frac{1 - 2^{-(b-c)(L+1)/2}}{1 - 2^{-(b-c)/2}} N + M_c \frac{2^{c(L+1)} - 1}{2^c - 1} & \text{if } c \neq b; \\ M_c (L+1) N + M_c \frac{2^{c(L+1)} - 1}{2^c - 1} & \text{if } c = b. \end{cases}
$$

- When $c < b$, setting $N = 2M_b(1 - 2^{-(b-c)/2})^{-1}\epsilon^{-1}\mu^{-1}$ guarantees that $V(v^{\text{V}-\text{MLMC}}) \leq \frac{\epsilon\mu}{2}$. Selecting the smallest $T^* \in \mathbb{N}$ such that the requirement on $T$ holds, then the total cost is bounded by

$$C \leq T^* C_{\text{iter}}^{\text{V}-\text{MLMC}} \leq T^*[2M_b M_c(1 - 2^{-(b-c)/2})^{-2}\mu^{-1}\epsilon^{-1} + M_c 2^c(2^c - 1)^{-1}(4M_a)^{c/a}\epsilon^{-c/a}].$$

- When $c = b$, setting $N = 2M_b(L+1)\epsilon^{-1}\mu^{-1}$, guarantees that $V(v^{\text{V}-\text{MLMC}}) \leq \frac{\epsilon\mu}{2}$. Selecting the smallest $T^* \in \mathbb{N}$ such that the requirement on $T$ holds, then the total cost is bounded by

$$C \leq T^* C_{\text{iter}}^{\text{V}-\text{MLMC}} \leq T^*[2M_b M_c(L+1)^2\mu^{-1}\epsilon^{-1} + M_c 2^c(2^c - 1)^{-1}(4M_a)^{c/a}\epsilon^{-c/a}].$$

Plugging in $L = \lceil 1/a \log(4M_a/\epsilon)\rceil$ we obtain the desired rate.

- When $c > b$, setting $N = 2M_b 2^{-(b-c)(L+1)/2}(2^{-(b-c)/2} - 1)^{-1}\mu^{-1}\epsilon^{-1}$ guarantees that $V(v^{\text{V}-\text{MLMC}}) \leq \frac{\epsilon\mu}{2}$. Selecting the smallest $T^* \in \mathbb{N}$ such that the requirement on $T$ holds, then the total cost is bounded by

$$C \leq T^*[2M_b M_c(1 - 2^{-(b-c)/2})^{-2}2^{-(b-c)}(4M_a)^{(c-b)/a}\mu^{-1}\epsilon^{-1-(c-b)/a} + M_c 2^c(2^c - 1)^{-1}(4M_a)^{c/a}\epsilon^{-c/a}],$$

where we uses $2^{(c-b)L} = \mathcal{O}(\epsilon^{-(c-b)/a})$ if $c > b$.

**Convex case**  By Lemma A.1(b), with stepsizes $\gamma_t = \frac{1}{2S_F}$ we have

$$\mathbb{E}[F^L(\widehat{x}_T) - F^L(x^L)] \leq \frac{V(v^{\text{V}-\text{MLMC}})}{2S_F} + \frac{2S_F\|x_1 - x^L\|_2^2}{T}.$$

To make sure that $\widehat{x}_T$ is $\epsilon/2$-optimal to $F^L$ so that $\widehat{x}_T$ is $\epsilon$-optimal to $F$, it suffices to have

$$T \geq 8S_F\|x_1 - x^L\|_2^2\epsilon^{-1};$$

$$V(v^{\text{V}-\text{MLMC}}) \leq \frac{\epsilon S_F}{2}.$$

Selecting $T^* \in \mathbb{N}$ the smallest $T$ satisfying the condition on iteration. By a similar argument in the strongly convex case, we set

$$N = \begin{cases} 2M_b(1 - 2^{-(b-c)/2})^{-1}\epsilon^{-1}S_F^{-1} & \text{if } c < b; \\ 2M_b(L+1)\epsilon^{-1}S_F^{-1} & \text{if } c = b; \\ 2M_b 2^{-(b-c)(L+1)/2}(2^{-(b-c)/2} - 1)^{-1}\epsilon^{-1}S_F^{-1} & \text{if } c > b. \end{cases}$$

It guarantees that $V(v^{\text{V}-\text{MLMC}}) \leq \epsilon S_F/2$. Plugging $N$ into Lemma B.2 for V-MLMC, we know that the total cost is bounded by

$$C = T^* C_{\text{iter}}^{\text{V}-\text{MLMC}} = \lceil 8S_F\|x_1 - x^L\|_2^2\epsilon^{-1}\rceil C_{\text{iter}}^{\text{V}-\text{MLMC}}$$

$$\leq \begin{cases} 16S_F\|x_1 - x^L\|_2^2\epsilon^{-1}(2M_b M_c(1 - 2^{-(b-c)/2})^{-2}\epsilon^{-1}S_F^{-1} + M_c 2^c(2^c - 1)^{-1}(4M_a)^{c/a}\epsilon^{-c/a}) & \text{if } c < b; \\ 16S_F\|x_1 - x^L\|_2^2\epsilon^{-1}(8M_b M_c a^{-2}\log^2(4M_a/\epsilon)\epsilon^{-1}S_F^{-1} + M_c 2^c(2^c - 1)^{-1}(4M_a)^{c/a}\epsilon^{-c/a}) & \text{if } c = b; \\ 16S_F\|x_1 - x^L\|_2^2\epsilon^{-1}(2M_b M_c(1 - 2^{-(b-c)/2})^{-2}2^{-(b-c)}(4M_a)^{(c-b)/a}\epsilon^{-1-(c-b)/a}S_F^{-1} \\ + M_c 2^c(2^c - 1)^{-1}(4M_a)^{c/a}\epsilon^{-c/a}) & \text{if } c > b. \end{cases}$$

Note that $(L+1)^2 \leq 4L^2 = 4a^{-2}\log^2(4M_a/\epsilon)$, $2^{(c-b)L} = \mathcal{O}(\epsilon^{-(c-b)/a})$ if $c > b$.

**Nonconvex smooth case**  By (21) and Lemma A.1(d), with stepsizes $\gamma_t = \frac{1}{S_F}$, we only need to select $N$ so that

$$\mathbb{E}\|\nabla F^L(\widehat{x}_T)\|_2^2 \leq \frac{2S_F(F^L(x_1) - F^L(x^L))}{T} + V(v^{\text{V}-\text{MLMC}}) \leq \frac{\epsilon^2}{4}.$$

It suffices to have

$$T \geq 16 S_F(F^L(x_1) - F^L(x^L))\epsilon^{-2};$$

$$V(v^{\mathrm{V-MLMC}}) \leq \frac{\epsilon^2}{8}.$$

Selecting the smallest $T^* \in \mathbb{N}$ such that the requirement on iteration is satisfied. To make sure that the variance is bounded by $\epsilon^2/4$, we set

$$N = \begin{cases} 8M_b(1 - 2^{-(b-c)/2})^{-1}\epsilon^{-2} & \text{if } c < b; \\ 8M_b(L+1)\epsilon^{-2} & \text{if } c = b; \\ 8M_b 2^{-(b-c)(L+1)/2}(2^{-(b-c)/2} - 1)^{-1}\epsilon^{-2} & \text{if } c > b. \end{cases}$$

Thus the total cost is bounded by

$$C \leq \begin{cases} 32S_F(F^L(x_1) - F^L(x^L))\epsilon^{-2}(8M_b M_c(1 - 2^{-(b-c)/2})^{-2}\epsilon^{-2} + M_c 2^c(2^c - 1)^{-1}(4M_a)^{c/a}\epsilon^{-2c/a}) & \text{if } c < b; \\ 32S_F(F^L(x_1) - F^L(x^L))\epsilon^{-2}(32M_b M_c a^{-2}\log^2(4M_a\epsilon^{-1})\epsilon^{-1} + M_c 2^c(2^c - 1)^{-2}(4M_a)^{c/a}\epsilon^{-2c/a}) & \text{if } c = b; \\ 32S_F(F^L(x_1) - F^L(x^L))\epsilon^{-2}(8M_b M_c(1 - 2^{-(b-c)/2})^{-2}2^{-(b-c)}(4M_a)^{(c-b)/a}\epsilon^{-2-2(c-b)/a} + \\ M_c 2^c(2^c - 1)^{-1}(4M_a)^{c/a}\epsilon^{-2c/a}) & \text{if } c > b. \end{cases}$$

Note that $(L+1)^2 \leq 4L^2 = 4a^{-2}\log^2(4M_a/\epsilon)$.

$\square$

## C.4 Expected Total Cost of RU-MLMC and RR-MLMC

In this subsection, we consider the expected total cost of RU-MLMC (see (7)) and RR-MLMC (see (8)). Both algorithms are unbiased and are applicable only when $c < b$, namely, the increase rate of the cost to generate a gradient estimator per level is strictly smaller than the decrease rate of the variance of $\nabla H^l(x, \zeta^l)$.

**Theorem C.5.** *Let A denote RU-MLMC or RR-MLMC. Under Assumption 2.1(b)(c)(d) with $c < b$ and Assumption 2.2, setting $q_l = 2^{-(b+c)l/2}(1 - 2^{-(b+c)/2})$ in RU-MLMC and RR-MLMC, we have the following results.*

- *When $F(x)$ is $\mu$-strongly convex, setting $\gamma_t = \frac{1}{\mu(t+2S_F/\mu^2)}$, to guarantee that $x_T^A$ is an $\epsilon$-optimal solution of F, the expected total cost of RU-MLMC and RR-MLMC satisfy*

$$C = \mathcal{O}(M_c M_b \epsilon^{-1}).$$

- *When $F(x)$ is convex, set $\gamma_t = 1/\sqrt{V(v^A)T}$, to guarantee $\widehat{x}_T^A$ is an $\epsilon$-optimal solution of F, T is large enough so that $\gamma_t \leq 1/(2S_F)$. The expected total cost of RU-MLMC and RR-MLMC satisfy*

$$C = \mathcal{O}(M_b M_c \epsilon^{-2}).$$

- *When $F(x)$ is $S_F$-smooth, setting $\gamma_t = 1/\sqrt{V(v^A)T}$, to guarantee $\widehat{x}_T^A$ is an $\epsilon$-stationarity point of F, the iteration complexity T has to be large enough so that $\gamma_t < 1/S_F$. The expected total cost of RU-MLMC and RR-MLMC satisfy*

$$C = \mathcal{O}(M_b M_c \epsilon^{-4}).$$

*Proof.* Notice that both algorithms are unbiased. Thus Lemma A.1 can be directly applied.

**Strongly Convex $F$** By Lemma A.1 (a2), to make sure that $x_T$ is an $\epsilon$-optimal solution, it suffices to have

$$T \geq \frac{S_F \max\{V(v^A), \mu^2(1 + 2S_F^2/\mu^2)\|x_1 - x^*\|_2^2)\}}{\mu^2 \epsilon}.$$

Selecting the smallest $T^* \in \mathbb{N}$ satisfying the requirement, by Lemma B.2, we have the upper bounds on the expected total cost such that

$$C = T^* C_{\text{iter}}^A \leq \frac{2S_F \max\{V(v^A), \mu^2(1 + 2S_F^2/\mu^2)\|x_1 - x^*\|_2^2)\}}{\mu^2 \epsilon} C_{\text{iter}}^A.$$

Plugging in $V(v^A)$ and $C_{\text{iter}}^A$ from Lemma B.2 where $A$ refers to RU-MLMC and RR-MLMC, we have the desired results. The expected total cost of RU-MLMC satisfies

$$C \leq 2S_F M_c \frac{1 - 2^{-(b+c)/2}}{1 - 2^{-(b-c)/2}} \max\{M_b \frac{1}{(1 - 2^{-(b-c)/2})} \frac{1}{1 - 2^{-(b+c)/2}}, \mu^2(1 + 2S_F^2/\mu^2)\|x_1 - x^*\|_2^2)\} \mu^{-2} \epsilon^{-1};$$

The expected total cost of RR-MLMC can be upper bounded by

$$C \leq 2S_F \frac{M_c 2^c (1 - 2^{-(b+c)/2})}{(2^c - 1)(1 - 2^{-(b-c)/2})} \max\{\frac{M_b}{1 - 2^{-(b-c)/2}}, \mu^2(1 + 2S_F^2/\mu^2)\|x_1 - x^*\|_2^2)\} \mu^{-2} \epsilon^{-1}.$$

**Convex $F$** To apply Lemma A.1(b), it requires that

$$\gamma_t = \frac{1}{\sqrt{V(v^A)T}} \leq \frac{1}{2S_F}.$$

Notice that by Lemma B.2, $V(v^A) = \mathcal{O}(1)$ for both RU-MLMC and RR-MLMC when $c < b$. As a result, for $T$ large enough, the condition on the stepsize is satisfied. By Lemma A.1, to make sure that $\widehat{x}_T$ is an $\epsilon$-optimal solution, it suffices to have

$$T \geq \epsilon^{-2} V(v^A)(1 + \|x_1 - x^*\|_2^2)^2$$

Selecting the smallest $T^* \in \mathbb{N}$ satisfying the requirement, by Lemma B.2, we have the upper bounds on the expected total cost such that

$$C = T^* C_{\text{iter}}^A \leq 2C_{\text{iter}}^A V(v^A)(1 + \|x_1 - x^*\|_2^2)^2 \epsilon^{-2}.$$

Plugging in $V(v^A)$ and $C_{\text{iter}}^A$ from Lemma B.2 where $A$ refers to RU-MLMC and RR-MLMC, we have the desired results. Particularly, the expected total cost of RU-MLMC satisfies

$$C \leq 2M_b M_c (1 - 2^{-(b-c)/2})^{-2}(1 + \|x_1 - x^*\|_2^2)^2 \epsilon^{-2}.$$

The expected total cost of RR-MLMC can be upper bounded by

$$C \leq 2M_b M_c (1 + \|x_1 - x^*\|_2^2)^2 \frac{2^c(1 - 2^{-(b+c)/2})}{(2^c - 1)(1 - 2^{-(b-c)/2})^2} \epsilon^{-2}.$$

**Nonconvex Smooth $F$** Similar to the convex case, we verify that the stepsizes selections are well-defined for large enough $T$. By Lemma A.1(c), to make sure that $\mathbb{E}\|\nabla F(\widehat{x}_T)\|_2^2 \leq \epsilon^2$, it requires that

$$T \geq V(v^A)(2(F(x_1) - F(x^*)) + S_F)^2 \epsilon^{-4}.$$

Selecting the smallest $T^*$ that the inequality holds, we can upper bound the total cost satisfies that

$$\mathbb{E}C \leq 2C_{\text{iter}}^A V(v^A)(2(F(x_1) - F(x^*)) + S_F)^2 \epsilon^{-4}.$$

Plugging in $V(v^A)$ and $C_{\text{iter}}^A$ from Lemma B.2 where $A$ refers to RU-MLMC and RR-MLMC, we have the desired results. Particularly, the expected total cost of RU-MLMC satisfies

$$C \leq 2M_b M_c(2(F(x_0) - F(x_T)) + S_F)^2 (1 - 2^{-(b-c)/2})^{-2} \epsilon^{-4}.$$

The expected total cost of RR-MLMC can be upper bounded by

$$C \leq 2M_b M_c(2(F(x_0) - F(x_T)) + S_F)^2 \frac{2^c(1 - 2^{-(b+c)/2})}{(2^c - 1)(1 - 2^{-(b-c)/2})^2} \epsilon^{-4}.$$

$\square$

# D  Discussion on Assumptions

Note that the assumption $F(x)$ is $\mu$-strongly convex in Lemma A.1(a1) and (a2) can be relaxed to PL condition. Recall that if $F(x)$ satisfies PL condition with parameter $\mu$, then it holds that

$$\|\nabla F(x)\|_2^2 \geq 2\mu(F(x) - F(x^*)).$$

One can derive PL condition from strong convexity. Supposing that $F$ is $\mu$-strongly convex, we have

$$
\begin{aligned}
F(x^*) \geq & F(x) + \nabla F(x)^\top (x^* - x) + \frac{\mu}{2}\|x^* - x\|_2^2 \\
\geq & \min_{\bar{x}} F(x) + \nabla F(x)^\top (\bar{x} - x) + \frac{\mu}{2}\|\bar{x} - x\|_2^2 \\
= & F(x) - \frac{1}{2\mu}\|\nabla F(x)\|_2^2.
\end{aligned}
\tag{23}
$$

Rearranging terms, we have the PL condition:

$$\|\nabla F(x)\|_2^2 \geq 2\mu(F(x) - F(x^*)).$$

In proving Lemma A.1(a1), we only use PL condition. As for (a2), one can show a similar result of (a2) using only the PL condition. The following corollary rigorously summarizes the expected error of SGD under PL condition.

**Corollary D.1.** *Suppose that $F(x)$ is $S_F$-smooth on $\mathbb{R}^d$ and suppose that there exists a constant $V > 0$ such that*

$$\mathbb{E}v(x) = \nabla F(x), \mathbb{V}(v(x)) \leq V,$$

*where expectations are taken w.r.t the randomness in $v$. Let $x^*$ be a minimizer of $F(x)$ on $\mathbb{R}^d$. If $F(x)$ satisfies PL condition with parameter $\mu$, we have the following results.*

- *For fixed stepsizes $\gamma_t = \gamma \in (0, \frac{1}{S_F}]$, it holds that*

$$\mathbb{E}[F(x_T) - F(x^*)] \leq (1 - \gamma\mu)^{T-1}[F(x_1) - F(x^*)] + \frac{S_F\gamma V}{2\mu}.$$

- *For time-varying stepsizes $\gamma_t = \frac{2}{\mu(t + 2S_F/\mu - 1)}$, we have*

$$\mathbb{E}[F(x_T) - F(x^*)] \leq \frac{2\max\{S_F V, \mu^2(1+z)(F(x_1) - F(x^*))\}}{\mu^2(T+z)}.$$

*Proof.* The proof of fixed stepsizes is the same as the proof of Lemma A.1(a1). As for the time varying stepsizes, following (16), we have

$$\mathbb{E}[F(x_{t+1}) - F(x^*)] \leq (1 - \gamma_t\mu)\mathbb{E}[F(x_t) - F(x^*)] + \frac{S_F\gamma_t^2}{2}V. \tag{24}$$

Plugging in $\gamma_t = \frac{2}{\mu(t+z)}$ with $z = 2S_F/\mu - 1$ so that $\gamma_t \in (0, 1/S_F]$, by induction, we have

$$\mathbb{E}[F(x_T) - F(x^*)] \leq \frac{2\max\{S_F V, \mu^2(1+z)(F(x_1) - F(x^*))\}}{\mu^2(T+z)}.$$

$\square$

Since the expected error under PL condition stays the same, the (expected) total cost of L-SGD, V-MLMC, RT-MLMC, RU-MLMC, and RR-MLMC under PL condition remains the same as the (expected) total cost achieved under the strong convexity assumption. For a more detailed discussion on PL condition and strong convexity, see Karimi et al. [24].

### D.1 Substituteable Bias Assumption under PL Condition

Recall that in Assumption 2.1(a), we make assumptions on the bias of the function value estimator, while in Assumption 2.2, we make assumptions on the bias of the gradient estimator. In this subsection, we demonstrate that under PL condition, one can derive the same (in terms of dependence on $\epsilon$) total cost of L-SGD, V-MLMC, and RT-MLMC by replacing Assumption 2.1(a) with Assumption 2.2. We also show that Assumption 2.2 cannot replace Assumption 2.1(a) for convex objectives using a counter example. RU-MLMC and RR-MLMC are unbiased methods under Assumption 2.2 and do not need Assumption 2.1(a).

Let $x_T^A$ denote the $T$-th iteration of algorithm $A$ for $A$ being L-SGD, V-MLMC, and RT-MLMC. The key step of such replacement is to show that under Assumption 2.2, the expected error of algorithm $A$, i.e., $\mathbb{E}[F(x_T^A) - F(x^*)]$, has a similar error decomposition like (9).

**Proposition D.1.** *Suppose that $F$ is $S_F$ smooth and satisfies the PL condition with parameter $\mu$, under Assumptions 2.1(b)(c) and Assumption 2.2, the followings hold.*

- *When using fixed stepsizes $\gamma_t = \gamma \in (0, 1/S_F]$, we have*

$$\mathbb{E}[F(x_T^A) - F(x^*)] \leq (1 - \gamma\mu)^{T-1}[F(x_1) - F(x^*)] + \frac{1}{2\mu}M_a 2^{-aL} + \frac{S_F\gamma}{2\mu}V(v^A). \quad (25)$$

- *When using time-varying stepsizes $\gamma_t = \frac{2}{\mu(t+2S_F/\mu-1)}$, we have*

$$\mathbb{E}[F(x_t^A) - F(x^*)] \leq \frac{2\max\{S_F V, \mu^2(1+z)(F(x_1) - F(x^*))\}}{\mu^2(t+z)} + \frac{1}{2\mu}M_a 2^{-aL}. \quad (26)$$

*Proof.* By smoothness of $F$, using $\gamma_t \leq 1/S_F$, and taking expectation conditioned on $x_t$, we have

$$\mathbb{E}F(x_{t+1}^A)$$
$$\leq F(x_t^A) + \nabla F(x_t^A)^\top \mathbb{E}(x_{t+1}^A - x_t^A) + \frac{S_F}{2}\mathbb{E}\|x_{t+1}^A - x_t^A\|_2^2$$
$$= F(x_t^A) + \nabla F(x_t^A)^\top \mathbb{E}(x_{t+1}^A - x_t^A) + \frac{S_F\gamma_t^2}{2}\mathbb{E}\|v(x_t^A)\|_2^2$$
$$= F(x_t^A) - \frac{\gamma_t}{2}(2\nabla F(x_t^A)^\top \mathbb{E}v(x_t^A)) + \frac{S_F\gamma_t^2}{2}\|\mathbb{E}v(x_t^A)\|_2^2 + \frac{S_F\gamma_t^2}{2}\text{Var}(v(x_t^A))$$
$$\leq F(x_t^A) + \frac{\gamma_t}{2}(-2\nabla F(x_t^A)^\top \mathbb{E}v(x_t^A) + \|\mathbb{E}v(x_t^A)\|_2^2) + \frac{S_F\gamma_t^2}{2}\text{Var}(v(x_t^A))$$
$$= F(x_t^A) + \frac{\gamma_t}{2}(-\|\nabla F(x_t^A)\|_2^2 + \|\mathbb{E}v(x_t^A) - \nabla F(x_t^A)\|_2^2) + \frac{S_F\gamma_t^2}{2}\text{Var}(v(x_t^A))$$
$$\leq F(x_t^A) - \frac{\gamma_t}{2}\|\nabla F(x_t^A)\|_2^2 + \frac{\gamma_t}{2}B_L + \frac{S_F\gamma_t^2}{2}V(v^A),$$

where the first inequality uses smoothness, the second inequality uses $\gamma_t \leq 1/S_F$, the third inequality uses Assumption 2.2, the last equality uses $\|a - b\|_2^2 = \|a\|_2^2 - 2a^\top b + \|b\|_2^2$. Using PL condition and taking full expectation, we have

$$\mathbb{E}[F(x_{t+1}^A) - F(x^*)]$$
$$\leq \mathbb{E}[F(x_t^A) - F(x^*)] - \frac{\gamma_t}{2}2\mu\mathbb{E}(F(x_t^A) - F(x^*)) + \frac{\gamma_t}{2}B_L + \frac{S_F\gamma_t^2}{2}V(v^A) \quad (27)$$
$$= (1 - \gamma_t\mu)\mathbb{E}(F(x_t^A) - F(x^*)) + \frac{\gamma_t}{2}B_L + \frac{S_F\gamma_t^2}{2}V(v^A).$$

- *If using fixed stepsizes, plugging in $\gamma_t = \gamma \in (0, 1/S_F]$, by induction, we have*

$$\mathbb{E}[F(x_T^A) - F(x^*)] \leq (1 - \gamma\mu)^{T-1}[F(x_1) - F(x^*)] + \frac{1}{2\mu}B_L + \frac{S_F\gamma}{2\mu}V(v^A)$$
$$\leq (1 - \gamma\mu)^{T-1}[F(x_1) - F(x^*)] + \frac{1}{2\mu}M_a 2^{-aL} + \frac{S_F\gamma}{2\mu}V(v^A).$$

- If using time varying stepsizes, plugging $\gamma_t = \frac{2}{\mu(t+z)} \in (0, 1/S_F]$ with $z = 2S_F/\mu - 1$ into (27), by induction, we have

$$\mathbb{E}[F(x_t^A) - F(x^*)] \leq \frac{2\max\{S_F V, \mu^2(1+z)(F(x_1) - F(x^*))\}}{\mu^2(t+z)} + \frac{1}{2\mu}B_L$$

$$\leq \frac{2\max\{S_F V, \mu^2(1+z)(F(x_1) - F(x^*))\}}{\mu^2(t+z)} + \frac{1}{2\mu}M_a 2^{-aL}.$$

$\square$

In the previous analysis of V-MLMC, we use fixed stepsizes. Particularly, we use Assumption 2.1, equation (9), and Lemma A.1(a1) to get

$$\mathbb{E}[F(x_T^A) - F(x^*)] \leq 2B_L + \mathbb{E}F^L(x_T^A) - F^L(x^L)$$

$$\leq 2M_a 2^{-aL} + (1 - \gamma\mu)^{T-1}[F^L(x_1) - F^L(x^L)] + \frac{S_F \gamma V(v^A)}{2\mu}. \tag{28}$$

Comparing (25) and (28), the only differences are in the constants, which do not affect the rate of the total cost.

In the previous analysis of L-SGD and RT-MLMC, we use time-varying stepsizes. Particularly, combining Assumption 2.1(a), equation (9), and Lemma A.1(a2), we have

$$\mathbb{E}[F(x_T^A) - F(x^*)] \leq 2B_L + \mathbb{E}F^L(x_T^A) - F^L(x^L)$$

$$\leq 2M_a 2^{-aL} + \frac{S_F \max\{V, \mu^2(1 + 2S_F^2/\mu^2)\|x_1 - x^*\|_2^2\}}{\mu^2(T + 2S_F^2/\mu^2)}. \tag{29}$$

Comparing (26) and (29), the only differences are in the constants, which do not affect the rate of the total cost.

*Remark* D.1. Unlike Assumption 2.1(a), Assumption 2.2 is not sufficient for obtaining global optimality gap when solving unconstrained optimization with convex objective $F$ using biased gradient methods. Suppose that Assumption 2.2 holds and that one finds an $\epsilon/4$-stationarity point of $F^L$ via some biased methods. The point is an $\epsilon$-stationarity point of $F$ by Assumption 2.2 for certain $L$. However, there is no link between the gradient norm of the point and the optimality gap for unconstrained optimization with convex objectives. In fact, one can show that for any $\epsilon \in (0, 1)$, there exists a convex smooth function $F : \mathbb{R}^d \to \mathbb{R}$ and a point $x_0$ such that $\|\nabla F(x_0)\|_2^2 = \epsilon^2$ and $F(x_0) - \inf_x F(x) > 1$. One example is the Huber function defined in the following and $x_0 \in \mathbb{R}^d$ is such that $\|x_0\|_2 = 2/\epsilon > \epsilon$.

$$F^{\text{Huber}}(x) = \begin{cases} \frac{1}{2}\|x\|_2^2 & \text{if } \|x\|_2 < \epsilon, \\ \epsilon(\|x\|_2 - \frac{\epsilon}{2}) & \text{if } \|x\|_2 \geq \epsilon. \end{cases}$$

One can easily see that $\|\nabla F(x_0)\|_2^2 = \epsilon^2$ but $F(x_0) - \inf_x F(x) = 2 - \frac{\epsilon^2}{2} > 1$. When encountering such functions, the biased gradient methods do not guarantee to converge to an $\epsilon$-optimal solution under Assumption 2.2.

# E   Applications to CSO Problems

We demonstrate the proof for Proposition 4.1. Note that Hu et al. [20] already proved the smoothness and the bias part. We focus on the variance of the oracle $\mathcal{SO}^l$.

*Proof.* We first consider the variance of $h^l(x, \zeta^l)$.

$$\mathbb{V}(h^l(x, \zeta^l))$$
$$\leq \mathbb{E}\|h^l(x, \zeta^l)\|_2^2$$
$$= \mathbb{E}\|\nabla\widehat{g}_{1:2^l}(x, \zeta^l)^\top \nabla f_{\xi^l}(\widehat{g}_{1:2^l}(x, \zeta^l))\|_2^2$$

$$\leq \mathbb{E}[\|\nabla \widehat{g}_{1:2^l}(x,\zeta^l)\|_2^2 \|\nabla f_{\xi^l}(\widehat{g}_{1:2^l}(x,\zeta^l))\|_2^2]$$
$$\leq L_f^2 L_g^2.$$

Next we show the variance of $H^l(x,\zeta^l)$. Notice that

$$\widehat{g}_{1:2^l}^l(x,\xi^l) = \frac{1}{2}[\widehat{g}_{1:2^{l-1}}^l(x,\xi^l) + \widehat{g}_{1+2^{l-1}:2^l}^l(x,\xi^l)].$$

$$\mathbb{V}(H^l(x,\zeta^l)) \leq \mathbb{E}\|H^l(x,\zeta^l)\|_2^2$$
$$=\mathbb{E}\|\nabla \widehat{g}_{1:2^l}^l(x,\xi^l)^\top \nabla f_{\xi^l}(\widehat{g}_{1:2^l}^l(x,\xi^l)) - \frac{1}{2}\nabla \widehat{g}_{1:2^{l-1}}^l(x,\xi^l)^\top \nabla f_{\xi_i^l}(\widehat{g}_{1:2^{l-1}}^l(x,\xi^l))$$
$$-\frac{1}{2}\nabla \widehat{g}_{2^{l-1}+1:2^l}^l(x,\xi^l)^\top \nabla f_{\xi_i^l}(\widehat{g}_{2^{l-1}+1:2^l}^l(x,\xi^l))\|_2^2$$
$$=\mathbb{E}\left\| \frac{1}{2}\nabla \widehat{g}_{1:2^{l-1}}^l(x,\xi^l)^\top \left[ \nabla f_{\xi^l}(\widehat{g}_{1:2^l}^l(x,\xi^l)) - \nabla f_{\xi^l}(\widehat{g}_{1:2^{l-1}}^l(x,\xi^l)) \right] \right.$$
$$\left. +\frac{1}{2}\nabla \widehat{g}_{2^{l-1}+1:2^l}^l(x,\xi^l)^\top \left[ \nabla f_{\xi^l}(\widehat{g}_{1:2^l}^l(x,\xi^l)) - \nabla f_{\xi^l}(\widehat{g}_{2^{l-1}+1:2^l}^l(x,\xi^l)) \right] \right\|_2^2$$
$$\leq \frac{L_g^2}{2}\mathbb{E}\left\| \nabla f_{\xi_i^l}(\widehat{g}_{1:2^l}^l(x,\xi^l)) - \nabla f_{\xi_i^l}(\widehat{g}_{1:2^{l-1}}^l(x,\xi^l)) \right\|_2^2$$
$$+\frac{L_g^2}{2}\mathbb{E}\left\| \nabla f_{\xi^l}(\widehat{g}_{1:2^l}^l(x,\xi^l)) - \nabla f_{\xi^l}(\widehat{g}_{2^{l-1}+1:2^l}^l(x,\xi^l)) \right\|_2^2$$
$$\leq \frac{L_g^2 S_f^2}{2}\left[ \mathbb{E}\left\| \widehat{g}_{1:2^l}^l(x,\xi^l) - \widehat{g}_{1:2^{l-1}}^l(x,\xi^l) \right\|_2^2 + \mathbb{E}\left\| \widehat{g}_{1:2^l}^l(x,\xi^l) - \widehat{g}_{2^{l-1}+1:2^l}^l(x,\xi^l) \right\|_2^2 \right]$$
$$=\frac{L_g^2 S_f^2}{4}\mathbb{E}\left\| \widehat{g}_{2^{l-1}+1:2^l}^l(x,\xi^l) - \widehat{g}_{1:2^{l-1}}^l(x,\xi^l) \right\|_2^2$$
$$\leq \frac{L_g^2 S_f^2}{4}\frac{2\sigma_g^2}{2^{l-1}}$$
$$=\frac{L_g^2 S_f^2 \sigma_g^2}{2^l},$$

where the last inequality uses the fact that $\widehat{g}_{2^{l-1}+1:2^l}^l(x,\xi^l)$ and $\widehat{g}_{1:2^{l-1}}^l(x,\xi^l)$ are independently identical distributed for a given $\xi^l$. $\qquad\square$

Corollary 4.1 is obtained via directly applying $a = b = c = 1$ in Theorem 3.2 for V-MLMC, Theorem 3.1 for RT-MLMC. In Table 3, the BSGD in Hu et al. [20] achieved $\widetilde{\mathcal{O}}(\epsilon^{-2})$, $\mathcal{O}(\epsilon^{-3})$, and $\mathcal{O}(\epsilon^{-6})$ sample complexity in the strongly convex, convex and nonconvex smooth setting, respectively. Since BSGD is a special case of the $L$-SGD framework in our paper, when plugging $a = b = c = 1$ into Theorem C.2, we directly recover the corresponding sample complexity. Hu et al. [20] additionally considered the variance reduction technique in the nonconvex smooth setting and achieves $\mathcal{O}(\epsilon^{-5})$ sample complexity, which is still worse than the V-MLMC and RT-MLMC in terms of the sample complexity.

# F   Numerical Experiments Details

The experiments are conducted on a personal computer with an Intel i7 CPU and GTX 2070 GPU.

## F.1   Synthetic Problem with Biased Oracles

We consider a synthetic quadratic programming problem.

$$\min_{x\in\mathbb{R}^d} F(x) := \frac{1}{2}\|x - z_\infty\|^2,$$

where $z_\infty = \lim_{n\to\infty} z_n$ and stochastic observation of $z_n$ can be obtained via some simulation process with cost $n$. Thus the approximation function is

$$F^l(x) = \frac{1}{2}\|x - z_{2^l}\|^2.$$

Let $z_n = (1 + \text{bias}/n)\mathbf{1}_d$ and $\widehat{z}_n$ be the output of the simulation such that $\widehat{z}_n \sim \mathbf{N}(z_n, \sigma^2\mathbb{I}_d)$, where $\mathbf{1}_d$ denotes a $d$ dimensional all one vector, $\mathbf{N}$ denote normal distribution with mean $z_n$ and covariance matrix $\sigma^2\mathbb{I}_d$. Here bias is a hyperparameter that controls the bias. One can construct $h^l$

$$h^l(x) = x - \widehat{z}_{2^l}, \text{ for } l = 0, \ldots, \infty;$$

and construct $H^l$

$$H^0(x) = x - \widehat{z}_{2^0}; \quad H^l(x) = -\widehat{z}_{2^l} + \frac{1}{2}(\widehat{z}'_{2^{l-1}} + \widehat{z}''_{2^{l-1}}) \text{ for } l = 1, \ldots, \infty$$

where $\widehat{z}'_{2^l}$ and $\widehat{z}''_{2^{l-1}}$ should be perfectly correlated with $\widehat{z}_{2^l}$. Without loss of generality, we construct $H^l(x) = \frac{1}{2^l}\sum_{i=1}^{2^l}\xi_i^l$ such that

$$\xi_i^l \sim \mathbf{N}(-z_{2^l} + z_{2^{l-1}}, \sigma^2\mathbb{I}_d) \text{ for } i = 1, \ldots, 2^l, \text{ for } l \geq 1.$$

$$H^l(x) \sim \mathbf{N}(-z_{2^l} + z_{2^{l-1}}, \sigma^2 2^{-2l} * \mathbb{I}_d) \text{ for } l \geq 1; \quad H^0(x) = x - \widehat{z}_{2^0}.$$

For three biased methods, V-MLMC, RT-MLMC, and LSGD, we test truncation level $L \in \{0, \ldots, 10\}$. For V-MLMC, we consider a mini-batch size $n_l = 2^{L-l}$ so that there would not be any extra costs incurred by rounding to integer numbers. For three randomized methods RT-MLMC, RU-MLMC, RR-MLC, we test geometry distribution with parameter $p \in \{0.1, \ldots, 0.9\}$. In the experiments, we set dimension $d$ as 100, bias as 1, variance $\sigma^2$ as 1, total budget as $4e + 4$. Note that the variance of $H^l$ decays exponentially with $b = 2$. The stepsize $\gamma$ is selected in $\{0.1, 0.01, 0.001, 0.0005, 0.0001\}$. We measure the performance of different algorithms via the average total costs over 10 trials. In each trial, $x_1$ are initialized according to $\mathbf{N}(5 * \mathbf{1}_d, \mathbb{I}_d)$. We do not specify any random seed in the experiments.

The best parameter setup of each method for synthetic quadratic programming is summarized in Figure 2. Note that each line in the figure reflects a trial of the methods with a different random initialization. The performance of each method is measured by the average error, which is reflected on top of each subfigure.

We summarized the implications from the figures as follows:

- On the synthetic quadratic programming example, all four MLMC gradient methods outperform the naive biased method LSGD.

- V-MLMC has the smallest variance as suggested by the theory. As a result, the stepsize of V-MLMC is much larger than the stepsizes of randomized MLMC gradient methods, which aligns with the theory. Although, in theory, the variance of RT-MLMC, RU-MLMC, RR-MLMC, and LSGD are all treated as $\mathcal{O}(1)$, in practice, we notice that the variances of RT-MLMC, RU-MLMC, and RR-MLMC are larger than the variance of LSGD due to the extra randomness introduced by sampling a level.

- Although in theory, V-MLMC could have extra costs incurred by rounding mini-batch sizes to integers numbers, in practice, we realize that one can always select a $N$ so that each mini-batch $n_l$ is an integer to avoid such extra cost. It implies potential improvements on the theory of V-MLMC.

- Comparing LSGD and RT-MLMC with large truncation level 10, one can immediately see that LSGD runs out of budget very fast and may not converge due to large per-iteration cost whereas RT-MLMC with truncation level 10 with proper stepsize can always converge. It aligns with the theory that the cost to construct RT-MLMC gradient estimators is much smaller than the cost to construct LSGD gradient estimator.

- For RR-MLMC and RU-MLMC, when the parameter $p$ of the geometry distribution is small, there is a high probability to generate a large level $l$, which can run out of budget very fast. However, for RT-MLMC, since the largest truncation level is restricted to 10, RT-MLMC performs robustly with small $p$. The numerical observation aligns with the theory of RU-MLMC, RR-MLMC, and RT-MLMC. When the parameter $p$ is large, i.e., close to 1, then with a high probability, the geometry distribution would generate a small level and would generate a very large level with a very small probability. When such rare events happen, there would be a sudden jump in the error. as we observed when $p = 0.8$ or $p = 0.9$.

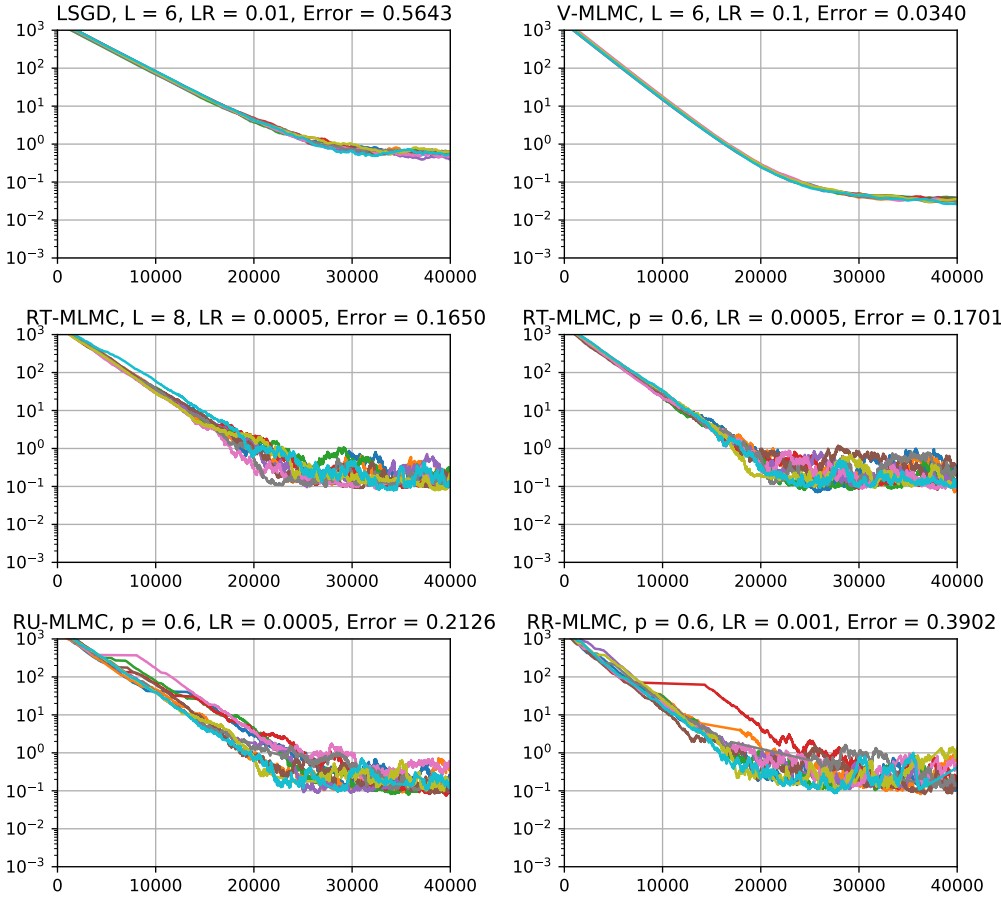

Figure 2: Best Setup of Different Algorithm on synthetic quadratic programming. The second row demonstrates the performance of RT-MLMC using the best selection of $L \in \{0, \ldots, 10\}$ for a given $p = 0.6$ on the left, and RT-MLMC using best selection of $p \in \{0.1, \ldots, 0.9\}$ for a given $L = 6$ on the right.

### F.2 Invariant Robust Regression

In this subsection, we conduct numerical experiments on an invariant robust regression [8] of the form:

$$\min_{x \in \mathbb{R}^d} \mathbb{E}_{\xi=(a,b)} f(\mathbb{E}_{\eta|a} \eta^\top w - b),$$

where $f$ is a loss function, $\xi = (a, b)$ represents the feature label pair, $\eta$ represents the poisoned feature. Such problem is a special case of conditional stochastic optimization. For $f$, we consider absolute loss and square loss. For simplicity, we consider $\eta = X + c$ where $c \sim \mathbf{N}(0, \sigma^2 \mathbb{I}_d)$.

In the experiments of absolute loss, we set dimension $d$ as 20, variance $\sigma^2$ as 1. We generate $N = 2000$ samples of $\xi$, i.e., 2000 feature label pairs such that $a \sim \mathbf{N}(\mathbf{1}_d, \mathbb{I}_d)$ and $b = a^\top x_0$ with $x_0 \sim \mathbf{N}(5 * \mathbf{1}_d, 10 * \mathbb{I}_d)$. Here $\mathbf{1}_d$ denotes a $d$-dimensional all one vector. The total budget is $1e + 5$. The MLMC component $H^l$ is constructed via (13). For all methods, we use a mini-batch of 50, i.e., at each iteration, we take 50 pairs of perturbed feature and label pairs. For V-MLMC, we use mini-batch size $m_l = 2^{L-l}$ for each level $l$. For RT-MLMC, RU-MLMC, and RR-MLMC, we additionally uses a mini-batch of 20 on the level generated by the geometry distribution to control their variance so that they can use larger stepsizes. Correspondingly, the cost is enlarged as well.

The best parameter setup of each method for invariant absolute regression, i.e., $f$ is the absolute loss, are summarized in Figure 3. The best parameter setup of each method for invariant least square, i.e., $f$ is the square loss, are summarized in Figure 4. Note that each line in the figure reflects a trial of the methods with a different random initialization. We performs 5 trials. The budget for invariant

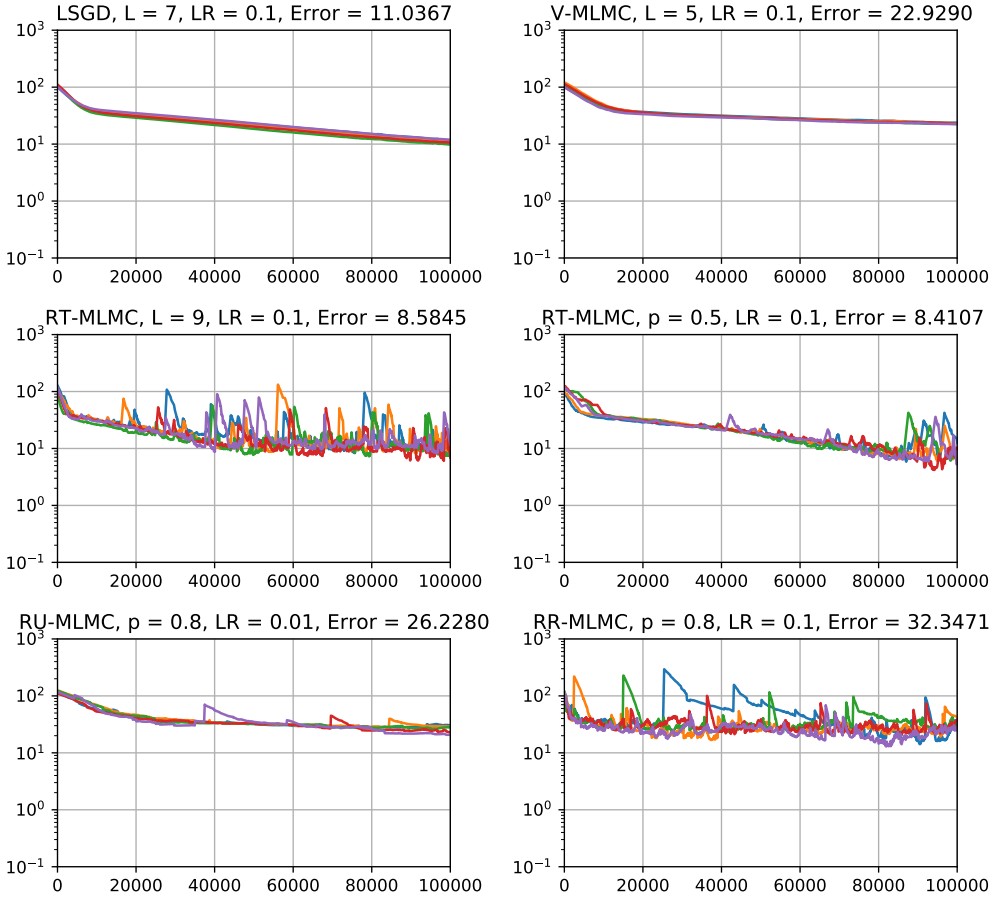

Figure 3: Best Setup of Different Algorithm on invariant absolute regression. The second row demonstrates the performance of RT-MLMC using the best selection of $L \in \{0, \ldots, 10\}$ for a given $p = 0.6$ on the left, and RT-MLMC using best selection of $p \in \{0.1, \ldots, 0.9\}$ for a given $L = 6$ on the right.

absolute regression is $1e + 5$ and for invariant least square is $4e + 4$. The performance of each method is measured by the average error, which is reflected on top of each subfigure.

We summarize the implications from invariant robust regression problems as follows. Note that the bias limit denotes the smallest average error that a biased method with a certain truncation level $L$, i.e., LSGD, V-MLMC, RT-MLMC, can achieve even if the budget is infinity. Note that as truncation level $L$ gets larger, the bias limit gets closer to $0$.

- In invariant absolute regression, we notice that RT-MLMC is better than LSGD and all other MLMC gradient methods achieve comparable performance. The major benefit of RT-MLMC is that it has a much smaller cost compared to LSGD. Thus when LSGD achieves the bias limit of truncation level 7, RT-MLMC can achieve the bias limit of a higher truncation level 9. Comparing RT-MLMC and unbiased MLMC gradient methods, RT-MLMC generally has a smaller variance. V-MLMC cannot beat LSGD since the cost of V-MLMC is much higher than LSGD. Thus with this sample budget, V-MLMC can only reach the bias limit of truncation level 5.

- For invariant least square, the gradient estimators could have larger variances. One can see that the optimal stepsizes selection is smaller in invariant least-square than that in the invariant absolute regression. Furthermore, to balance the variance of RT-MLMC, it has to use a smaller stepsize compared to LSGD. As a result, using $1e + 5$ budgets, RT-MLMC has not converged to the bias limit of a large truncation level yet. On the other hand, V-MLMC has a much larger per-iteration cost. As variance grows, the number of iterations of V-MLMC grows. Thus $1e + 5$

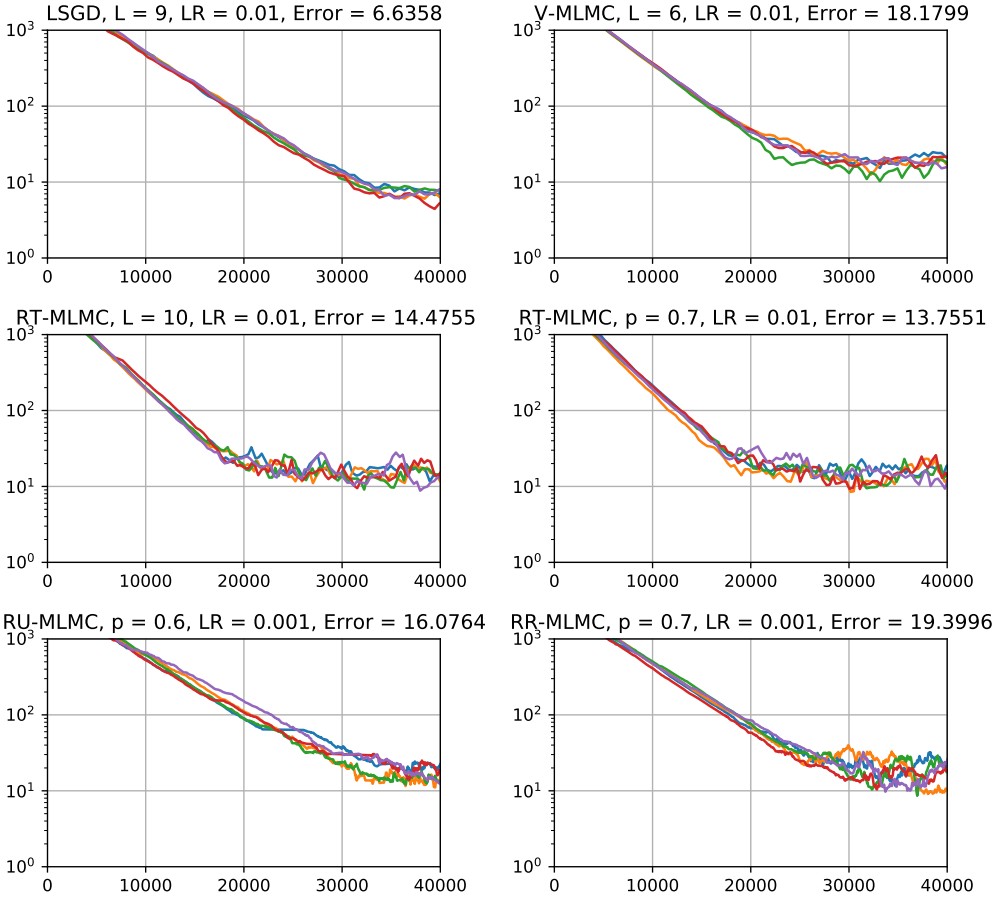

Figure 4: Best Setup of Different Algorithm on invariant least square. The second row demonstrates the performance of RT-MLMC using the best selection of $L \in \{0, \dots, 10\}$ for a given $p = 0.6$ on the left, and RT-MLMC using best selection of $p \in \{0.1, \dots, 0.9\}$ for a given $L = 6$ on the right.

budget for V-MLMC is not enough to reach the bias limit as well. We further test the variance times the cost to construct gradient estimators. Such value of MLMC gradient methods is similar to that of LSGD. Thus MLMC gradient methods do not achieve better performance. It means that the constants that we hide in $\mathcal{O}(1)$ in theory also play an important role in the practical performance of MLMC gradient methods.

- Although in theory mini-batch reduces the variance, but it increases the cost of the gradient estimator, there is no influence on the eventual cost if the optimal stepsize is used. In practice, however, we do not know the optimal stepsizes. Mini-batches allow larger stepsizes and stabilize the training process.