# OpenReview forum: "On the Bias-Variance-Cost Tradeoff of Stochastic Optimization"
_NeurIPS.cc/2021/Conference — NeurIPS 2021 Poster_

### Official Review · Reviewer_Jayx · 2021-07-13

**Rating:** 6
**Confidence:** 2

**Summary:**

This paper provides a systematic study of the bias-variance-cost tradeoff for several MLMC gradient methods and L-SGD, under the assumption that (i) there exist a sequence of obj. approximations; (ii)  there exist stochastic oracles which generate unbiased estimates of the gradient for the obj. approximations, and unbiased estimates of the difference between function approximation gradients. They assume three parameters a, b, c which respectively describe the bias, variance of the stochastic oracle, and cost to query the oracle, and provide insights on which class of algorithms work better under different configurations of a, b, c. They also show that for conditional stochastic optimization problems, MLMC methods can significantly reduce the sample complexity.

**Limitations And Societal Impact:**

No. I do not think it is related to any negative societal impact.

**Main Review:**

Overall this paper is well-written, and provides useful insights for practitioners regarding which MLMC method to use for real-world problems. The illustration on the CSO problem also demonstrates the value of MLMC. One question I have is that, in practice, since F(x) is unknown, how do you get an estimate of a, in order to know which is the proper MLMC method? I assume b and c can be estimated in some way since they only involve the function approximations, not the underlying true function.

On Page 5 there seem to be some typos: L-SGD estimator: ... for n_L times to obtain {h^L}, not gradient of h? Same error also exists in Eq. (4); V-MLMC: ... to obtain {H}, not gradient of H.

It would also be helpful to provide some experimental comparisons of these MLMC methods, to (i) validate your theoretical results, and (ii) give convincing empirical evidence of the superiority and limitations of different methods under different situations.

I personally expect to see the application of MLMC methods in min-max optimization, which you mentioned as part of the future directions.

**Time Spent Reviewing:**

4

---

> ### Author Response · Authors · 2021-08-10
> **Response to Reviewer Jayx**
>
> We thank Reviewer Jayx for the positive comments. Below we address the questions brought up by the reviewer.
>
> **Q1**: In practice, since $F(x)$ is unknown, how do you get an estimate of $a$, in order to know which is the proper
> MLMC method? I assume $b$ and $c$ can be estimated in some way since they only involve the function approximations,
> not the underlying true function.
>
> **Response**:
> For structured problems like CSO and DRO, we do know the formulation of $F$. In such cases, it is hard to evaluate $F$ and its gradient at a given point $x$. However, we can approximate the true objective using the structural information and analyze theoretically what are $a,b,c$. When we do not even know structural information of $F$, it might be impossible to construct approximation functions $\\{F^l\\}$ at first hand. It might be an interesting question how one can solve an optimization problem with limited information on the structure. We leave it for future investigation as it is beyond the scope of the current paper.
>
> Note that the value of $a$ is only used to determine the total number of levels $L$ in these algorithms. In practice, we can treat $L$ as a hyperparameter and run a grid search to obtain the best $L$. This applies to $b,c$ as well.
>
>
>
> **Q2**:  Typos on Page 5  : L-SGD estimator: ... for $n_L$ times to obtain ${h^L}$, not gradient of $h$? Same error
> also exists in Eq. (4); V-MLMC: ... to obtain ${H}$, not gradient of $H$
>
> **Response**:  Thanks for pointing it out. It should be $h^L$  and $H^l$ rather than $\nabla h^L$ and $\nabla H^l$ in lines 159-162.
>
>
> **Q3**: It would also be helpful to provide some experimental comparisons of these MLMC methods, to (i) validate your
> theoretical results, and (ii) give convincing empirical evidence of the superiority and limitations of different methods
> under different situations.
>
> **Response**:
> Please refer to the experiments in the Letter to All Reviewers.
>
> **Q4**:
> Application of MLMC methods in min-max optimization.
>
> **Response**:
> We have tried to analyze MLMC for nonconvex-strongly concave stochastic min-max optimization after we submitted the paper. We observed that MLMC provides a comparable yet not superior complexity bound of order $\mathcal{O}(\epsilon^{-4})$: the sample complexity of MLMC matches with the results from [Lin et al, 2020, Chen et al, 2020] and the lower bounds for nonconvex stochastic optimization [Arjevani et al, 2019]. It remains interesting and open whether MLMC can improve the complexity bounds for stochastic min-max problems in other settings.
>
>
> Suppose one intends to solve a stochastic min-max problem with strongly-concave objective over $y$, i.e., $\min_x \max_y \mathbb{E}_\xi f_\xi(x,y)$. At iteration $t$ with $x_t$, generate one sample $\xi_t$ from distribution of $\xi$ and generate a level $l$ from the truncated geometric distribution. One can construct the building block $H^l(x,\zeta^l)$ of MLMC gradient estimators via the following procedure.
>
> * Generate a sample path $\zeta^l = \\{\xi_{tj}\\}_{j=1}^{2^l}$.
>
> * Perform stochastic gradient ascent on $\max_y \mathbb{E}_\xi f_\xi(x_t,y)$ for $2^{l-1}$ iterations using the first half of the sample path $\zeta^l$ with a fixed stepsize to obtain $y^{l-1}$; then update another $2^{l-1}$ iterations using the same stepsize and the second half of the sample path to obtain $y^{l}$.
>
> * Construct $H^l(x_t,\zeta^l)= \nabla f_\xi(x_t,y^l)-\nabla f_\xi(x_t,y^{l-1})$.
>
> * Construct MLMC gradient estimator as we mentioned in Section 2.

---

> > ### Author Response · Authors · 2021-08-12
> > **Reference to Reviewer Jayx**
> >
> > Reference:
> >
> > Chen, Tianyi, Yuejiao Sun, and Wotao Yin. "Tighter Analysis of Alternating Stochastic Gradient Method for Stochastic Nested Problems." arXiv preprint arXiv:2106.13781 (2021).
> >
> > Yossi Arjevani, Yair Carmon, John C Duchi, Dylan J Foster, Nathan Srebro, and Blake Wood-
> > 354 worth. Lower bounds for non-convex stochastic optimization. arXiv preprint arXiv:1912.02365,
> > 355 2019.
> >
> > Tianyi Lin, Chi Jin, and Michael Jordan. On gradient descent ascent for nonconvex-concave minimax problems. In International Conference on Machine Learning, pages 6083–6093.
> > 409 PMLR, 2020.

---

### Official Review · Reviewer_voJd · 2021-07-15

**Rating:** 6
**Confidence:** 3

**Summary:**

This paper consider stochastic optimization when we only have access to a sequence of biased functions that converge to the true function. They apply various Multilevel Monte-Carlo methods (MLMC) to handle the bias-variance tradeoff. Convergence rate and computational complexity are computed for both convex and non-convex objective. They also demonstrate the superiority of MLMC gradient methods when applied to conditional stochastic optimization.

**Limitations And Societal Impact:**

Yes.

**Main Review:**

The setting that one only have access to a sequence of approximations to the objective function is well motivated with some important applications like conditional stochastic optimization and distributionally robust optimization, so the problem that this paper studied is of interest to the optimization community. The idea of introducing MLMC to gradient based optimization is not new [1], so the main contribution of this paper is not the algorithms but the theoretical analysis of convergence rate and computational complexity. They make systematical convergence and complexity analysis for both convex and non-convex objectives with clear dependency on the assumptions parameters, which is novel to this area to the best of my knowledge. Although I didn’t fully check the detail proofs, the paper is in-general well-written and reasonably easy to follow. Here are some points that I would like to hear the response from the authors:

1. Numerical example is missing in Section 4. Sometimes the MLMC based method may involved a huge constant term in either the convergence rate or the complexity (especially the MLMC method that involve randomized level), so it would be better to have some numerical results to confirm the practical usefulness.

2. Consider the conditional stochastic optimization problem, say min_x E[f(E[g(Y, x)|Z])]. With some smooth conditions on f and g, we are actually able to come up with unbiased estimator of E[f(E[g(Y, x)|Z])] or its gradient by randomized MLMC [2], because they are of the form h(E[W]). So perhaps we don’t really need to follow the current setting (sequence of approximated functions) to solve the problem?

[1]: Jose Blanchet, Donald Goldfarb, Garud Iyengar, Fengpei Li, and Chaoxu Zhou. Unbiased simulation for optimizing stochastic function compositions. arXiv preprint arXiv:1711.07564, 2017.
[2]: Jose H Blanchet and Peter W Glynn. Unbiased monte carlo for optimization and functions of expectations via multi-level randomization. In 2015 Winter Simulation Conference (WSC), pages 3656–3667. IEEE, 2015.

**Time Spent Reviewing:**

3.5

---

> ### Author Response · Authors · 2021-08-10
> **Response to Reviewer voJd**
>
> We thank the Reviewer voJd for the feedback. Below we address the questions.
>
> **Q1**: Numerical example is missing in Section 4.
>
> **Response**:  Please refer to the experiments results in the Letter to All Reviewers.
>
> **Q2**:  Consider the conditional stochastic optimization problem, say $min_x E[f(E[g(Y, x)|Z])]$. With some smooth
> conditions on $f$ and $g$, we are actually able to come up with unbiased estimator of $E[f(E[g(Y, x)|Z])]$ or its gradient by
> randomized MLMC by Blanchet and Glynn, 2015, because they are of the form $h(E[W])$. So perhaps we don’t really need to follow the current
> setting (sequence of approximated functions) to solve the problem?
>
> **Response**:
> We agree with the reviewer that if there are other ways of constructing an unbiased gradient estimator, such sequence of approximation function might not be necessary. However, building unbiased estimators is not easy for many general problems and is the core of many statistics and machine learning research. Particularly, we would like to point out sometimes, even if one can construct unbiased gradient estimators, it might not be optimal. For instance,  the randomized MLMC in Blanchet and Glynn 2015 is a way to construct unbiased function value estimators with finite variance for conditional stochastic optimization (CSO). However, when using the method to construct an unbiased gradient estimator, it coincides with the RU-MLMC method in our paper. Although it is unbiased, either the variance or the expected cost of the gradient estimator will be infinite by Lemma 5, Remark 7 in Page 17, and Line 299 that $b=c$ for CSO. As a result, one needs to turn to biased truncated MLMC methods like RT-MLMC and V-MLMC.

---

### Official Review · Reviewer_FGif · 2021-07-17

**Rating:** 7
**Confidence:** 3

**Summary:**

The paper studies a family of MLMC methods for gradient estimation and systematically analyses their performances under different stochastic optimization settings (strongly convex, convex, and nonconvex). In each setting, the order of the total cost is given in terms of the decay rates of the bias and variance, and the corresponding increasing rates of the oracle cost. It is shown in the CSO problems that some of the MLMC methods have lower computational costs compared to existing state-of-the-art (e.g., [19]).

**Limitations And Societal Impact:**

Please see the comments and concerns in the quality and clarity parts above.

**Main Review:**

Originality: The paper provides a comprehensive and systematic study on the computational costs of MLMC gradient descent methods, which is not available in the literature as far as I know and presents a good contribution.

Quality: The theoretical claims are well supported with detailed proofs.

It would be good if the authors can provide examples where b<c and b>c, corresponding to table 1. This would give stronger evidence that each regime is worth studying. For example, in Section 1.1, the DRO example, can you provide the values of a,b and c and which regimes it could correspond to in table 1?

It seems that there is a sharp phase transition for the performance of RU-MLMC and RR-MLMC from c<b to c>=b, as seen in table 1. When c<b, they are the best. But when c>=b, they are not applicable at all. This seems to suggest some important considerations on the constant in the big-O notation that depends on these parameters. Discussions on this issue would be helpful.

Clarity: The paper is well organized and the main results are presented clearly. But some arguments could be explained more carefully:

In Theorem 1, it is not very clear to me why n_L should be chosen in the stated way. In remark 4, it is explained that this is a result of “optimizing the total cost with respect to n_L in the continuous space”, but I think the author(s) could provide a more concrete argument. Similarly, there should be some discussion of the choice of q_L in Theorem 2.

Line 240-241: If my understanding is correct, it seems that if 1>c/a, then V-MLMC is O(\epsilon^{-c/a}) better instead of O(\epsilon^{-1}) better then L-SGD (comparing line 240 and remark 3).

Table 2: in the last column of the row for RU-MLMC, should n_l be q_l instead?

Significance: The study is theoretically significant and insightful in guiding the choice of MLMC gradient schemes. It also points to approaches that can outperform state-of-the-art methods.

**Time Spent Reviewing:**

2 hours

---

> ### Author Response · Authors · 2021-08-10
> **Response to Reviewer FGif**
>
> We thank Reviewer FGif for the positive feedback and suggestions. Below we address the questions brought by the reviewer.
>
> **Q1**:
> It would be good if the authors can provide examples where $b<c$ and $b>c$, corresponding to table 1. This would give
> stronger evidence that each regime is worth studying. For example, in Section 1.1, the DRO example, can you provide
> the values of a,b and c and which regimes it could correspond to in table 1?
>
> **Response**:
> Thanks for the suggestion. Below we list some examples for different regimes of $b,c$. Some are discussed explicitly in the paper; others can be easily derived, which we will include in our updated version.
>
> * Case $b=c$: Examples include
>     * CVaR DRO problem [25]: $a=2$, $b=1$, $c=1$;
>     * $\chi^2$-penalized DRO problem [25]:   $a=1$, $b=1$, $c=1$
>     * CSO problem with smooth inner and outer functions (Section 4): $a=1, b=1, c=1$
> * Case $b>c$: Consider the CSO problem such that 1) $\nabla f_\xi$ is smooth and 2) $g_\eta(\cdot,\xi)$ is linear and sub-Gaussian for any given $\xi$. One can easily verify that in this case $a = 1, b=2, c=1$.
> * Case $b<c$: An interesting example that we recently found is the so-called contextual stochastic optimization [Bertsimas and Nathan, 2020], which has the form
>     $$ \min_x F(x,\xi) := \mathbb{E}_{\eta} [f(x;\eta)|\xi]. $$
>
> The objective can be approximated by a sequence of functions based on the $k$-nearest neighbors:
>
> $$ F^l(x,\xi):=\mathbb{E} \sum_{i=1}^{m_l}v_{1:m_l}^{k_l}(\xi,\xi_i)f(x,\eta_i), $$
>
> where expectation is taken over $ \zeta^l = ( \xi_i, \eta_i)_{i=1}^{m_l} $,
>
> and $v_{a:b}^k(\xi,\xi_i)=\frac{1}{k}\mathbb{I}$ ($\xi_i$ is within the $k$-nearest neighbors of $\xi$ among $\\{\xi_i\\}_{a}^{b}$).
>
> Under some regularity assumptions, we are able to show that $\mathbb{E}[F^l(x,\xi)-F(x,\xi)]=\mathcal{O}(\frac{k_l}{m_l})^{1/d_\xi}$. Setting $k_l=2^l, m_l=2^l(d_\xi/2+1)$, we have $a=1,b=1,c=d_\xi/2+1$. In this case, $b<c$.
>
>
>
>
> **Q2**:  It seems that there is a sharp phase transition for the performance of RU-MLMC and RR-MLMC from $c<b$ to $c>=b$, as
> seen in table 1. When $c<b$, they are the best. But when $c>=b$, they are not applicable at all. This seems to suggest some
> important considerations on the constant in the big-O notation that depends on these parameters. Discussions on this
> issue would be helpful.
>
>
> **Response**:
> Thanks for the question. Indeed, the big-O notation of the per-iteration cost and the variance of RU-MLMC and RR-MLMC should involve the following quantity (see Lemma 5 in page 17):
>
> $$ R^\infty(\frac{c-b}{2}) = \lim_{L\rightarrow\infty} \sum_{l=0}^L 2^{(c-b)l/2}. $$
>
> When $c\geq b$, $R^\infty(\frac{c-b}{2})$ goes to infinity.
>
>
>
> **Q3**: In Theorem 1, it is not very clear to me why $n_L$ should be chosen in the stated way. In remark 4, it is explained that this
> is a result of “optimizing the total cost with respect to $n_l$ in the continuous space”, but I think the author(s) could
> provide a more concrete argument. Similarly, there should be some discussion of the choice of $q_l$ in Theorem 2.
>
> **Response**:
> For the choice of $\\{q_l\\}_{l=0}^L$ ($L=\infty$ for RU-MLMC and RR-MLMC), we discuss it in Remark 8 on Page 18.
>
> Finding the optimal choice of $\\{q_l\\}$ can be  formulated an optimization problem:  minimizing the total cost $ (\sum_{l=0}^L q_l C_l)(\sum_{l=0}^L V_l/q_l ) $ subject to $ \sum_{l=0}^L q_l=1$ and $q_l\geq 0 $. The optimal solution corresponds to the proposed $ q_l\propto 2^{-(b+c)l/2} $.  We will add discussion on the choice of $q_l$ after Theorem 2.
>
> As for the optimal choice of $\\{n_l\\}_{l=0}^L$, this would reduce to solving an integer programming to ensure the batch size $\\{n_l\\}$ are integers.
>
>  Here we consider a relaxation by setting $n_l= q_l\cdot n$, where $\sum_{l=0}^L q_l=1$ and $q_l\geq 0$.  As a result, we have $n_l\propto 2^{-(b+c)l/2} n$. Given that $n_l$ is the mini-batch size, we set $n_l = \lceil 2^{-(b+c)l/2} n\rceil$.
>
> **Q4 Minor Issues:**
> 1. Line 240-241: If my understanding is correct, it seems that if $1>c/a$, then V-MLMC is $\mathcal{O}(\epsilon^{-c/a})$ better instead of
> $\mathcal{O}(\epsilon^{-1})$ better then L-SGD (comparing line 240 and remark 3).
> 2. Table 2: in the last column of the row for RU-MLMC, should $n_l$ be $q_l$ instead?
>
> **Response:** Thanks for pointing out the typo.
> 1. In line 240-241, it should be V-MLMC is always $\mathcal{O}(\epsilon^{-\min\\{1,c/a\\}})$ better than L-SGD.
> 2. It is a typo. It should be $q_l$ in the last column of the row for RU-MLMC.
>
> **Reference**:
> Bertsimas, Dimitris, and Nathan Kallus. "From predictive to prescriptive analytics." Management Science 66.3 (2020): 1025-1044.

---

### Official Review · Reviewer_Q47Z · 2021-07-19

**Rating:** 5
**Confidence:** 4

**Summary:**

This paper describes various MLMC debiasing schemes for computing stochastic gradient estimates, and applies them to conditional stochastic optimization problems. Here, unlike other more-standard problems in optimization it is impossible to compute true gradients of the objective function: the objective is an expected value over infinitely many points. However, as this paper observes, approximate minimizers to these problems may nevertheless be computed. The main ideas underlying this result are the use of sampling and a multi-level Monte Carlo (MLMC) to generate low-bias stochastic gradient estimates: these are then combined with stochastic gradient descent.

**Limitations And Societal Impact:**

-

**Main Review:**

Overall, the paper studies a very well-studied problem and achieves improved theoretical guarantees for it. I found the analysis highly technical and difficult to follow, but the results seem correct to me. The analyses of four different schemes for bias reduction may be helpful for future work on related problems, although the reason for including these is unclear (as the parameters a,b,c in the paper have specified values in the application to CSO)

However, the paper's result seems to essentially derive from combining SGD with standard MLMC-based gradient estimation schemes (for example, the ones in [4]). The utility of biased oracles in stochastic methods has been observed several times before. While some of the bias-reduced estimators given here are new, the bias and variance analyses of these estimators follows a standard template laid out in previous work [4,25] (indeed, the calculations of Lemmas 4 and 5 may be obtained by slight modifications of the calculations in Appendix C of [25]).

In addition, while the application to CSO is compelling I did not completely understand how the results compare to those of [19]. In particular, the paper claims a result that contradicts a lower bound of [19] and justifies this by stating that the oracle of [19] is more restricted than the setup considered here. Some more description of this behavior and a justification of the stronger access model presented here would be helpful .

**Time Spent Reviewing:**

2

---

> ### Author Response · Authors · 2021-08-10
> **Response to Reviewer Q47Z**
>
> We thank Reviewer Q47Z for the feedback. Below we clarify the reviewer's concerns.
>
> **Q1**:
> The analyses of four different schemes for bias reduction may be helpful for future work on related problems, although the reason for including these is unclear (as the parameters a,b,c in the paper have specified values in the application of CSO).
>
> **Response**:
>  The objective of this paper is to provide a systematic study of first-order methods for solving a broad class of stochastic optimization problems for which it is costly to obtain unbiased gradient estimators. We are particularly interested in understanding the bias, variance, and cost of different gradient estimators and how they affect the total sample complexity and computational complexity. The four schemes we analyze are the most natural schemes one would construct and clearly demonstrate the tradeoffs of the bias, variance, and cost through the parameters $a,b,c$.
>
>
>
>
> **Q2**:
> While some of the bias-reduced estimators given here are new, the bias and variance analyses of these estimators follow a standard template laid out in previous work [4,25].
>
> **Response**:
> As we point out in our response to your Q1, the objective of this paper is to provide a systematic study of first-order methods for solving a broad class of stochastic optimization problems for which it is costly to obtain unbiased gradient estimators. Even though the bias and variance analysis is standard, providing such a systematic study is valuable. Indeed, our analysis covers several natural ways of constructing MLMC estimators and clearly demonstrates the tradeoffs of the bias, variance, and cost through the parameters $a,b,c$ under strongly-convex, convex, and nonconvex smooth regimes. In contrast, [25] restricted to DRO under the convex setting using RT-MLMC; [4] focused on constructing unbiased MLMC estimation for objective rather than analyzing the cost for optimization problems. Particularly, the result in [4] that one can construct unbiased function estimation for $h(\mathbb{E}(Y))$ with finite variance does not generalize to the gradient estimation. As we have shown for the CSO problem, the unbiased gradient construction using RU-MLMC and RR-MLMC would lead to infinite variance and one has to resort to truncated MLMC methods.
>
>
> **Q3**:
> While the application to CSO is compelling, I did not completely understand how the results compare to those of [19]. The paper claims a result that contradicts a lower bound of [19] and justifies this by stating that the oracle of [19] is more restricted than the setup considered here. Some more descriptions of this behavior and a
> justification of the stronger access model presented here would be helpful.
>
> **Response**:
> In [19], they assumed the algorithm has access to a stochastic gradient oracle (black-box oracle) that returns a biased gradient of CSO with $\mathcal{O}(\epsilon)$ bias and variance $\mathcal{O}(1)$ at a query cost of $\mathcal{O}(\epsilon^{-1})$. In contrast, our paper shows that using RT-MLMC, one can construct a specific gradient estimator (white-box oracle) that has the same bias and variance $\\mathcal{O}(1)$ property, but the cost is of order $\mathcal{O}(\log(\epsilon^{-1}))$.  To summarize,
>
> * [19]: black-box oracle, bias $\mathcal{O}(\epsilon)$, variance $\mathcal{O}(1)$, cost $\mathcal{O}(\epsilon^{-1})$.
>
> * ours: white-box oracle, bias $\mathcal{O}(\epsilon)$, variance $\mathcal{O}(1)$, cost $\mathcal{O}(\log(\epsilon^{-1}))$.
>
> Since the iteration complexity (which only depends on the bias and the variance) stays the same,  the reduced per-iteration cost leads to a reduced total cost.

---

### Author Response · Authors · 2021-08-11
**Letter to All Reviewers: Experiments Results**

We are grateful for the detailed review and valuable suggestions.

After the submission, we conduct experiments on invariant logistic regression and invariant least square regression:

$$ \min_w \mathbb{E}_{\xi=(X,y)} [\log(1+\exp(-y^\top \mathbb{E}[\eta w|\xi])]; $$

$$
    \min_w \mathbb{E}_{\xi=(X,y)}\\|\mathbb{E}[\eta w|\xi]-y \\|^2;
$$

where the inner expectation is taken over the conditional distribution of $\eta | \xi$.

In the equation, $X$ is a random feature, $y$ is the corresponding label and $\eta$ is a random perturbed observation of the feature $X$, i.e., $\eta =X + \sigma$. We use the MNIST dataset and simulated data for feature $X$ and label $y$. Note that such invariant learning problems are special cases of CSO problems.  Unlike the general results shown in Section 4 that $a=b=c=1$, one can show that $a=1, b=2, c=1$ for such special cases if $\sigma$ is a sub-Gaussian random variable. For discussion on how to derive this, please refer to Q1 for the reviewer FGif. It implies that all four MLMC methods and L-SGD apply to such problems.

There are four hyperparameters in the experiments:

* Variance of perturbation $\sigma$. It corresponds to the bias of the approximation function as suggested by Lemma 4.1.

* Probability $p$ denotes the parameter of the (truncated) geometric distribution of levels in RT-MLMC, RU-MLMC, and RR-MLMC;

* Levels $L$ corresponds to the truncation level for L-SGD, V-MLMC, and RT-MLMC.


For the invariant least square regression problem, we use simulated data. The variance of perturbation $\sigma$ is $0.1$, $10\%$ of the variance of the simulated feature $X$. For each method, we find the best combination of probability and levels.  The following tables summarize the results under the high accuracy regime and the low accuracy regime. The performance in the high accuracy regime is measured by the training loss given $10^6$ samples, and the performance in the low accuracy regime is measured by the number of samples needed to reach a certain accuracy $0.02$.

* High accuracy regime: the goal is to achieve the smallest training loss using $10^6$ samples of $\sigma$.

| Methods  | Probability       |Levels     |Number of Epoch   | Training loss|
| :---        |    :----:   | :----:     | :----:      |          ---: |
| V-MLMC  | NA       | 2      | 1000|0.000198|
| RT-MLMC  | 0.9        | 4     | 1000| 0.000186|
|RU-MLMC | 0.9        | NA       | 1000|0.002425|
| RR-MLMC |0.9      | NA  |1000|0.001146|
| L-SGD  | NA       | 2      |1000|0.000193|

RT-MLMC achieves the best result in the high accuracy regime with a truncation level $4$. Particularly the decay of the error of RT-MLMC in terms of total sample used is much faster than L-SGD. (Figures will be added in the updated version) Such observation highlights the importance of MLMC methods in balancing the bias and the cost by properly selecting $L$ and $p$, as we observed in our theoretical results.   The optimal truncation levels for all truncated methods are either $2$ or $4$, which aligns with our analysis that $L=\mathcal{O}(\log(1/\epsilon))$ is not too large if the approximation error is small. The example shows that indeed MLMC methods are useful.

As for different MLMC methods, we further observe that 1) RR-MLMC method performs better than RU-MLMC method in practice, even though they are treated indifferently in our theoretical analysis; 2) V-MLMC method might not need a very large mini-batch in practice.


* Low accuracy regime: the goal is to use the smallest number of samples to reach $0.02$ loss.

|Methods  | Probability       |Levels  | Number of samples/1000|
| :---        |    :----:   | :----:     |         ---: |
| V-MLMC  | NA       | 1      | 9.73|
| V-MLMC  | NA       | 2     | 39.64|
| RT-MLMC  | 0.8        | 2      | 12.37|
|RU-MLMC | 0.8        | NA       | 13.61|
| RR-MLMC |0.8      | NA  |11.29|
| L-SGD  | NA       | 2      | 19.7|
| L-SGD/RT-MLMC  | NA       | 1      | 10.15|

To achieve a low accuracy, we observe that all truncation methods, V-MLMC, RT-MLMC, and L-SGD, achieve the best performance when $L=1$ comparing to $L>2$. (When $L=1$, RT-MLMC reduces to L-SGD, V-MLMC reduces to mini-batch L-SGD.) It suggests that for problems which do not require a high accuracy, L-SGD with a small truncation level could provide good approximated minimizers. It aligns with our analysis that the upper bounds on the optimality gap in strongly convex case is proportional to $\mathcal{O}$(bias + variance / iterations) by equation (19) and Lemma 2. If allowing a larger optimality gap, one only need small $L$ in L-SGD to control the bias. On the other hand, from $L=1$ to $L=2$, the total sample size of RT-MLMC grows slower than other truncated methods.



For the invariant logistic regression problem on MNIST dataset, we set $\mathrm{Var}(\sigma)/\mathrm{Var}(X)$ to be $10\%$, $50\%$, and $100\%$, corresponding to denote low bias, middle bias, and large bias regime, respectively (Lemma 4.1 shows that the variance of $\sigma$ relates to the bias of the approximation function). The results are summarized in the following table, where Var stands for variance, Pro stands for probability, Levels stands for truncation, Samples stands for the total sample size is 50*50000, V, RT, RU, RR, LSGD stands for methods.

| Var | Pro | Level | Samples   | V train |V test| RT train | RT test | RU train | RU test| RR train | RR test| LSGD train | LSGD test |
|  :---   | :----: |:----:| :----:|:----:|:----:|:----:| :----:|:----:|:----:|:----:| :----:|:----:| ---:|
|0.1|0.2|3|50|0.278827|0.9235|0.268517|0.9223|1.503121|0.6879|1.464305|0.6803|0.270394|0.924|
|0.1|0.8|3|50|0.278654|0.9232|0.251318|0.9249|0.362934|0.9073|0.248175|0.9278|0.271248|0.9228|
|0.1|0.5|10|50|1.430426|0.7835|0.28459|0.9202|0.558198|0.8757|0.271227|0.9274|0.751181|0.8514|
|0.1|0.8|10|50|1.430829|0.7823|0.25086|0.9272|0.360392|0.9081|0.247363|0.9263|0.752191|0.8421|
|0.5|0.5|5|50|0.353676|0.9104|0.273566|0.9247|0.487917|0.8845|0.279682|0.9222|0.310428|0.9187|
|1|0.5|5|50|0.40811|0.902|0.285967|0.9228|1.007754|0.8043|0.293055|0.9201|0.340467|0.9135|
|1|0.8|5|50|0.408235|0.9017|0.272815|0.9239|0.420862|0.8983|0.274845|0.9232|0.34092|0.9118|

* For fixed variance, the performance of RU-MLMC and RR-MLMC gets worse when $p$ decreases. It aligns with our analysis that with small $p$, unbiased MLMC methods can have a higher chance of generating large levels, leading to high expected per-iteration cost and high variance. It suggests that for implementation purposes, one should use large $p$ for unbiased randomized MLMC methods.
* For fixed levels $L$, we observe that the performance of RT-MLMC with proper probability $p$ is always better than L-SGD for $L\geq 3$. For $L=1$, RT-MLMC reduces to L-SGD. For $L=2$, we observe that RT-MLMC converges faster than LSGD to reach a moderate accuracy at the beginning of the training. However, sometimes LSGD would has smaller training loss and better validation accuracy.  We will add these figures in the updated version.  For fixed probability $p$ and level $L$, the performance of all methods gets worse as the variance increases. L-SGD, V-MLMC, and RR-MLMC are more robust against such change.
* In our experiments, we notice that V-MLMC does not require a very large mini-batch size to achieve good performance, and LSGD does not need very large $L$. The reason might be that the bias of the approximation function decays in order of $1/2^L$, which is very fast. [25] on DRO problems make similar observations on their $\chi^2$-penalized DRO problems.  It remains interesting to formally characterize the condition when MLMC would have significant benefits in practice.

---

### Decision · Program_Chairs · 2021-09-27

**Decision:**

Accept (Poster)

**Comment:**

The paper provides a systematic description and analysis of multilevel Monte-Carlo (MLMC) gradient estimators for stochastic convex optimization and describes application of these estimators to conditional stochastic optimization. Despite some reservations regarding the novelty of some of the developments in the paper, the reviewers appreciated its generality and agreed it could be of value to the NeurIPS community. Furthermore, the reviewers agreed that the experimental results outlined in the author response would be a useful combination to the paper.

In light of this, I recommend that the paper be accepted to NeurIPS and encourage the authors to carefully revise the paper to address the issues brought up during the review. In particular, the authors should include a comprehensive report of their experimental results, consulting the NeurIPS submission checklist to ensure that all the pertinent information is included. Moreover, in cases where experiments indicate the negative finding, that MLMC does not outperform standard bias SGD in practice, I encourage the authors to clearly and transparently highlight this conclusion, as it is as valuable to researchers as a positive finding.